# Differential Column Measurements Using Compact Solar-Tracking Spectrometers

Jia Chen[1,4], Camille Viatte[2], Jacob K. Hedelius[2], Taylor Jones[1], Jonathan E. Franklin[1], Harrison Parker[3], Elaine W. Gottlieb[1], Paul O. Wennberg[2], Manvendra K. Dubey[3], and Steven C. Wofsy[1]

[1]School of Engineering and Applied Sciences and Department of Earth and Planetary Sciences, Harvard University, Cambridge, MA 02138, USA
[2]Division of Geological and Planetary Sciences, California Institute of Technology, Pasadena, CA 91125, USA
[3]Earth and Environmental Sciences, Los Alamos National Laboratory, Los Alamos, NM 87545, USA
[4]now at Electrical and Computer Engineering, Technische Universität München, Munich, 80333, Germany

*Correspondence to:* Jia Chen (jia.chen@tum.de)

**Abstract.** We demonstrate the use of compact solar-tracking Fourier transform spectrometers (Bruker EM27/SUN) for differential measurements of the column-averaged dry-air mole fractions of $CH_4$ and $CO_2$ within urban areas. Using Allan variance analysis, we show that the differential column measurement has a precision of 0.01% for $X_{CO_2}$ and $X_{CH_4}$ using an optimum integration time of 10 min, corresponding to Allan deviations of 0.04 ppm, and 0.2 ppb, respectively. The sensor system is very stable over time and after relocation across the continent. We report tests of the differential column measurement, and its sensitivity to emission sources, by measuring the downwind-minus-upwind column difference $\Delta X_{CH_4}$ across dairy farms in the Chino area, California and using the data to verify emissions reported in the literature. Ratios of spatial column differences $\Delta X_{CH_4}/\Delta X_{CO_2}$ were observed across Pasadena within the Los Angeles basin, indicating values consistent with regional emission ratios from the literature. Our precise, rapid measurements allow us to determine significant short-term variations (5-10 minutes) of $X_{CO_2}$ and $X_{CH_4}$, and to show that they represent atmospheric phenomena.

Overall, this study helps establish a range of new applications for compact solar-viewing Fourier transform spectrometers. By accurately measuring the small differences in integrated column amounts across local and regional sources, we directly observe the mass loading of the atmosphere due to the influence of emissions in the intervening locale. The inference of the source strength is much more direct than inversion modeling using only surface concentrations, and less subject to errors associated with small-scale transport phenomena.

## 1 Introduction

Cities and their surrounding urban regions occupy less than 3% of the global land surface (Grimm et al. (2008)), but are home to 54% of the world population (WHO (2014)) and account for more than 70% of global fossil-fuel $CO_2$ emissions (Gurney et al. (2015)). Hence, accurate methods for measuring urban and regional scale carbon fluxes are required in order to design and implement policies for emission reduction initiatives.

It is challenging to use *in situ* measurements of $CO_2$ and $CH_4$ to derive emission fluxes in urban regions. Surface concentrations typically have high variance due to the influence of nearby sources, and they are strongly modulated by mesoscale transport phenomena that are difficult to simulate in atmospheric models. These include the variation of the depth of the planetary boundary layer (PBL), sea breeze, and topographic flows, etc. (McKain et al. (2012); Bréon et al. (2015)).

The mass loading of the atmosphere can be directly determined by measuring the column integrated amount of a tracer through the whole atmosphere. Column measurements are insensitive to vertical redistribution of tracer mass, e.g. due to growth of the PBL, and are also less influenced by nearby point sources whose emissions are concentrated in a thin layer near the surface. Column observations are more compatible with the scale of atmospheric models and hence provide stronger constraints for inverse modeling (Lindenmaier et al. (2014)).

One potential drawback, however, is that column observations are sensitive to surface emissions over a very wide range of spatial scales, spanning nearby emissions and all those upwind in the urban, continental, and hemispheric domains. In this paper we demonstrate how to use simultaneous measurements of the column-averaged dry-air mole fractions (DMFs) of $CH_4$ and $CO_2$ (denoted by $X_{CH_4}$ and $X_{CO_2}$, respectively) at upwind and downwind sites to mitigate this limitation. The horizontal gradients *within* a region are relatively insensitive to surface fluxes upwind of the domain, providing favorable input for regional

flux inversions.

We use three matched, compact Fourier transform spectrometers (FTS) to measure the small (0.1%) differences of $X_{CH_4}$ and $X_{CO_2}$, and we demonstrate sufficient precision and speed to determine emission rates at the urban scale. By directly measuring spatial and temporal gradients of the mass loading, we reduce the sensitivity of inverse model results to atmospheric fine structure, such as may arise from vertical redistribution of trace gases, and that often complicates the interpretation of surface

*in situ* data (Chang et al. (2014)).

Our ground-based network of spectrometers measuring gradients of column amounts could enable new approaches to validate the *urban-rural gradients* of satellite observations such as OCO-2 (Crisp et al. (2008); Frankenberg et al. (2015)) and TROPOMI (Veefkind et al. (2012)). In contrast to the large, high spectral resolution instruments of the Total Carbon Column Observing Network (TCCON), which are not easily relocated, the compact spectrometers can be deployed directly under satel-

lite tracks that pass near major cities, to assess potential artifacts in satellite-derived tracer gradients that might arise from urban or rural differences in aerosol burden, land surface properties, etc.

Several recent papers have studied column-averaged concentrations of trace gases to derive source fluxes. Wunch et al. (2009) observed diurnal patterns for $X_{CO_2}$, $X_{CH_4}$, and $X_{CO}$ over Los Angeles, similar to the model simulations of McKain et al. (2012) for Salt Lake City. Kort et al. (2012) used GOSAT satellite data to measure the difference between $CO_2$ columns

inside and outside Los Angeles, and to derive a top-down inventory for $CO_2$. Papers by Stremme et al. (2009, 2013) and Té et al. (2012) used total column measurements from a ground-based Fourier transform spectrometer (FTS) to estimate and monitor CO emission in Mexico City and Paris, respectively. Mellqvist et al. (2010) studied plumes from industrial complexes, and Lindenmaier et al. (2014) examined plumes from two power plants and discriminated them. Kort et al. (2014) quantified large methane sources missing in inventories at Four Corners, New Mexico. However, these studies did not have simultaneous

upwind and downwind column data, one of the novel elements of the present paper.

Frey et al. (2015) and Hase et al. (2015) reported deployments of multiple FTSs of the same type as employed here, deriving calibration and stability characteristics in a field setting. We extend this analysis by determining the Allan variances of column concentration *differences* between spectrometer pairs deployed side-by-side, providing a rigorous assessment of the precision of the differential column measurements.

Here we study local scale gradients in $X_{CO_2}$ and $X_{CH_4}$ in two applications. First, we deployed our spectrometers upwind and downwind of the dairy farms in Chino, California (about 50 km$^2$ area), and use the data to compare with emissions reported in the literature. A second application uses the observed ratio of differences in $X_{CO_2}$ and $X_{CH_4}$, i.e. $\Delta X_{CH_4}/\Delta X_{CO_2}$, to characterize emission ratios for these gases within the Los Angeles basin.

In another application of the compact spectrometers, we co-located spectrometers to demonstrate measurement of short-term (5-10 minutes) variations of column-averaged DMFs in the atmosphere. The high precision measurements with rapid scan rates are an advantage of the compact spectrometers compared to larger, higher spectral resolution spectrometers that have scan rates in the minute range. We show that high frequency observations can be used to quantify the influence of sporadic events, such as plumes, transient peaks, or instabilities across the top of the mixed layer (ML), on measurements in urban areas.

## 2 Differential Column Network

### 2.1 Column Measurement and Existing FTS Network

Solar-tracking FTSs can be used to measure the gas column number densities, i.e. the number of gas molecules per unit area in the atmospheric column (column$_G$, unit: molec. m$^{-2}$). The sun is used as light source and the FTS is located on the ground for measuring the solar radiation transmitted through the atmosphere. The recorded sun radiation spectrum is broadband and covers the absorption fingerprints of diverse gas species including $CO_2$, $CH_4$, $H_2O$ and $O_2$. The attenuation of the solar intensity at specific frequencies provides a measure for the column number density of various gases. For further details of modeling the atmospheric transmittance spectrum, please see Wunch et al. (2011) and Hase et al. (2004), for the working principles of FTS please refer to Davis et al. (2001), and Griffiths and De Haseth (2007).

The existing FTS networks include NDACC (Hannigan (2011)), i.e."Network for the Detection of Atmospheric Composition Change", and TCCON (Toon et al. (2009), Wunch et al. (2010), Wunch et al. (2011)). NDACC measures at mid-infrared wavelengths and detects atmospheric $O_3$, $HNO_3$, $HCl$, $HF$, $CO$, $N_2O$, $CH_4$, $HCN$, $C_2H_6$, and $ClONO_2$, chosen to help understand the physical and chemical state of the upper troposphere and the stratosphere. The TCCON network focuses on column measurements of greenhouse gases, mainly $CO_2$, $CH_4$, $N_2O$ and $CO$, at near-infrared wavelengths. It uses the Bruker IFS 125HR spectrometer that is large in dimension (container size) and heavyweight (>500 kg, Bruker (2006)). The spectra in the TCCON network are recorded with a spectral resolution of approx. 0.02 cm$^{-1}$ and require about 170 s for one forward/backward scan pair (Hedelius et al. (2016)).

## 2.2 Differential Column Measurement with Compact FTS

Our differential column network uses at least two spectrometers to make simultaneous measurements of column number densities of $CO_2$, $CH_4$, and $O_2$. We then compute the column-averaged DMFs (Wunch et al. (2011)) for each gas G, i.e.,

$$X_G = \frac{column_G}{column_{O_2}} \cdot 0.2095, \tag{1}$$

and differences, i.e.,

$$\Delta X_G = X_G^d - X_G^u, \tag{2}$$

where $X_G^d$ and $X_G^u$ stand for column-averaged DMFs at downwind and upwind sites.

Our sensors are two EM27/SUN FTS units owned by Harvard University, and one owned by Los Alamos National Laboratory, #45, 46, and 34 Bruker Optics (designated *ha*, *hb*, and *pl*, respectively). They are compact (62.5 cm $\times$ 35.6 cm $\times$ 47.3 cm) and lightweight (22.8 kg including the sun tracker), with a spectral resolution of 0.5 cm$^{-1}$ and a scan time of 5.8 s (forward or backward scan). The EM27/SUN tracks the sun precisely (1 $\sigma$: 11 arc s) using a camera for fine alignment of the tracking mirrors (Gisi et al. (2011)). It is mechanically very robust, with excellent precision in retrieving $X_{CO_2}$ and $X_{CH_4}$ (Gisi et al. (2012), Klappenbach et al. (2015), Hedelius et al. (2016)), comparable to Bruker IFS 125HR used in the TCCON network (Wunch et al. (2011)).

We carried out extensive side-by-side measurements of *ha* and *hb* in Boston and Pasadena, over many months, thoroughly examining precision and robustness, and also compared these systems to the TCCON spectrometer in Pasadena, California (Hedelius et al. (2016)). We confirm that these spectrometers are stable (Frey et al. (2015)). We show that comparing pairs of them cancels out most of the systematic error and bias from diverse sources, e.g. spectroscopic and retrieval errors, instrument bias, and errors in pressure and temperature, enabling us to determine 0.1% differences in column-averaged DMFs across the network.

## 3 System Characterization

### 3.1 Allan Analysis for System Precision

Known standards cannot be exchanged for the ambient air in a total column measurement, hence it is difficult to assess the precision of atmospheric measurements end-to-end. Two commonly used literature methods for precision estimates have been based on:

1. *Measurements of the standard deviation of the DMF time series, with the trend removed subtracting a moving average (Gisi et al. (2012)).* This approach is confounded by real variations in the atmosphere that occur on short time scales (*vide infra*).

2. *The residual of the spectral fit.* This estimate does not separate systematic errors, e.g. errors in spectroscopic database and modeling of instrument line shape, from the measurement noise, and therefore may overestimate the true random uncertainty of the measurement (*cf.* Fu et al. (2014)).

In this paper we use the Allan variance method (Allan (1966), Werle et al. (1993)) to estimate the measurement precision.
Fig. 1 shows the Allan deviations of the *differences* in column-averaged DMFs measured simultaneously by *ha* and *hb* at the same location, i.e, $\Delta X_G(t) = X_G^{hb}(t) - X_G^{ha}(t)$. The Allan variance of $\Delta X_G$ is denoted by $\sigma^2_{allan,\Delta X_G}$, which is the expectation value $\langle\rangle$ of the difference between adjacent samples averaged over the time period $\tau$:

$$\sigma^2_{allan,\Delta X_G}(\tau) = \frac{1}{2}\left\langle(\overline{\Delta X_{G,n+1}} - \overline{\Delta X_{G,n}})^2\right\rangle,\tag{3}$$

with $\overline{\Delta X_{G,n}} = \frac{1}{\tau}\int_{t_n}^{t_n+\tau}\Delta X_G(t)dt$. Practically $\overline{\Delta X_{G,n}}$ is the mean of all $\Delta X_G$ measurements within the time interval $[t_n, t_n+\tau)$.
According to the Allan variance plots:

- The optimal integration time, given by the minimum in the Allan deviation, is 10 to 20 min, for both $X_{CO_2}$ and $X_{CH_4}$.

- When averaging 10 min, the precision (1 Allan deviation) of the EM27/SUN differential column measurement is 0.04-0.05 ppm (0.01%) for $X_{CO_2}$ and 0.1-0.2 ppb (0.01%) for $X_{CH_4}$. Since the two instruments are statistically uncorrelated, the individual measurement noise is smaller by factor $1/\sqrt{2}$, indicating precision comparable to near infrared *in situ* laser spectrometers with commensurate optical path length and integration time (Picarro (2015a, b)). Note that these precision estimates represent the full end-to-end processing of the observations, including deriving the spectrum from the interferogram, retrieving the column number densities in the atmosphere, and normalizing with the $O_2$ column amount to obtain the column-averaged DMFs.

- When integrating less than 10 min, the Allan deviation follows a slope of -1/2 in the double logarithmic scale, indicating white noise ($\tau^{-1/2} \to f^0$), which has a constant power spectral density over the frequency $f$. As the averaging time $\tau$ increases beyond 10 min, the Allan deviation rises a little, showing a small color noise component ($\tau^{1/2} \to f^{-2}$), which arises from instrument drift, in part due to temperature differences inside of the spectrometers. There is also a small divergence between the measurements of *ha* and *hb* at high solar zenith angles, traceable to their slightly different instrument line shapes (ILSs). The measured ILS parameters are given in Appendix A. Microscale eddies have durations of 10 s to 10 min and length scales from tens to hundreds of meters (Stull (1988), Fig. 2.2). Therefore atmospheric turbulence probably does not play a major role in the Allan plot because there is little color noise within time scale $\leq 10$ min for two spectrometers looking along atmospheric paths separated by only a few meters.

We use a shorter integration time (5 min) for measuring emissions from local and regional scale sources (Sec. 4.1 and 4.2), in order to retain high frequency atmospheric signals, giving us precision of 0.05-0.06 ppm for $\Delta X_{CO_2}$ and 0.2-0.3 ppb for $\Delta X_{CH_4}$ (see Fig. 1). To study the short-term variations due to pollution plumes or turbulent eddies we use 2 min integration time (Sec. 4.3).

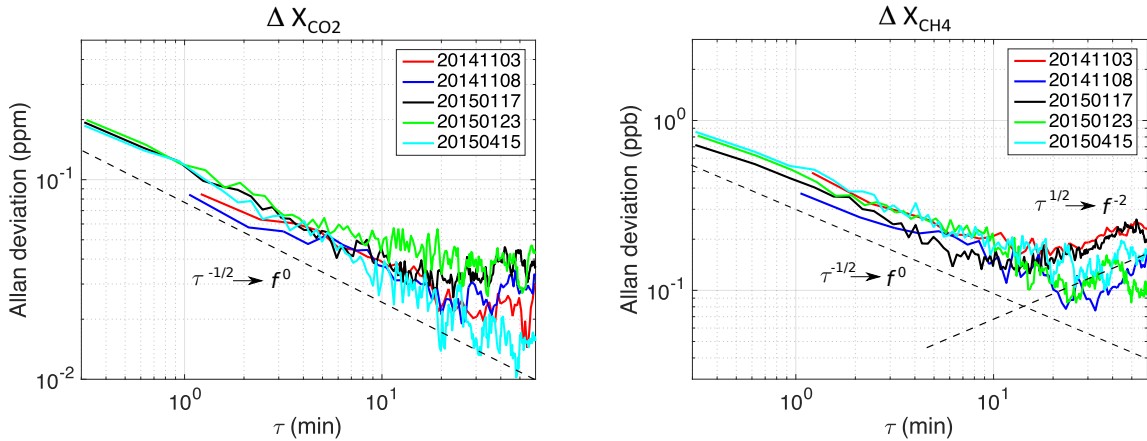

**Figure 1.** Allan deviation $\sigma_{allan,\Delta X_{CO2}}$ and $\sigma_{allan,\Delta X_{CH4}}$ as a function of the integrating time $\tau$. The black dashed lines represent a slope of $-1/2$ and a slope of $1/2$, which correspond to power spectral densities $S(f) = f^0$ (white noise) and $S(f) = f^{-2}$ (Brownian noise), respectively. The Allan deviation follows a slope of $-1/2$ up to an integration time of 10 to 20 min, and then stays constant ($S(f) = f^{-1}$), and subsequently turns over to a slope of $1/2$, which describes a linear drift.

## 3.2 System Stability

Differential column observations by two spectrometers will inevitably have bias in addition to fluctuations and drift. For the EM27/SUN, small differences in the alignments of the interferometers result in minute, but observable and systematic, deviations in the retrieval results. We examined the biases between $ha$ and $hb$ over a long period of time to determine if these

errors can be effectively corrected by applying a constant calibration factor to the retrieval of one instrument to match the performance of the other. The calibration factors are determined assuming a linear model, i.e. $X_G^{hb} = X_G^{ha} \cdot \overline{R_G}$, and for each gas individually.

The value of $\overline{R_G}$ was consistent over time for the two Harvard EM27/SUNs, including shipment across the contiguous United States (Fig. 2, Table 1). We used two retrieval software systems, I2S (interferogram-to-spectrum) combined with GFIT

nonlinear least-squares spectral fitting retrieval software (Wunch et al. (2015), Hedelius et al. (2016)), and PROFFIT (Hase et al. (2004)). The calibration factors are slightly different for GFIT and PROFFIT, traceable to their specific modeling of the ILS, various priors for the volume mixing ratio profiles, and unequal spectral microwindows that are used. Nevertheless, $\overline{R_G}$ is consistent in Boston and Pasadena, before and during the campaign, when the same retrieval settings are used. Retrievals for $ha$ have been scaled with $\overline{R_G}$ for the Allan analysis (Sec. 3.1) and for the scientific applications (Sec. 4) below. Calibration

factors for $pl$ are shown in Appendix B. The measured ILS parameters of the two Harvard EM27/SUNs are also consistent over time and across continent, as given in Appendix A.

|  | $\overline{R_{CH_4}}$ | | $\overline{R_{CO_2}}$ | |
| --- | --- | --- | --- | --- |
|  | GFIT | PROFFIT | GFIT | PROFFIT |
| Before | 0.99574 | 0.99813 | 0.99877 | 0.99838 |
| During | 0.99580 | 0.99809 | 0.99881 | 0.99834 |
| Both | 0.99578 | 0.99810 | 0.99880 | 0.99835 |

**Table 1.** Calibration factors $\overline{R_G}$ for $X_{CH_4}$ and $X_{CO_2}$ before and during the field campaign, determined by forcing a linear regression line to go through the origin. $\overline{R_G}$, determined using all data, are provided in the last row and used for the field study.

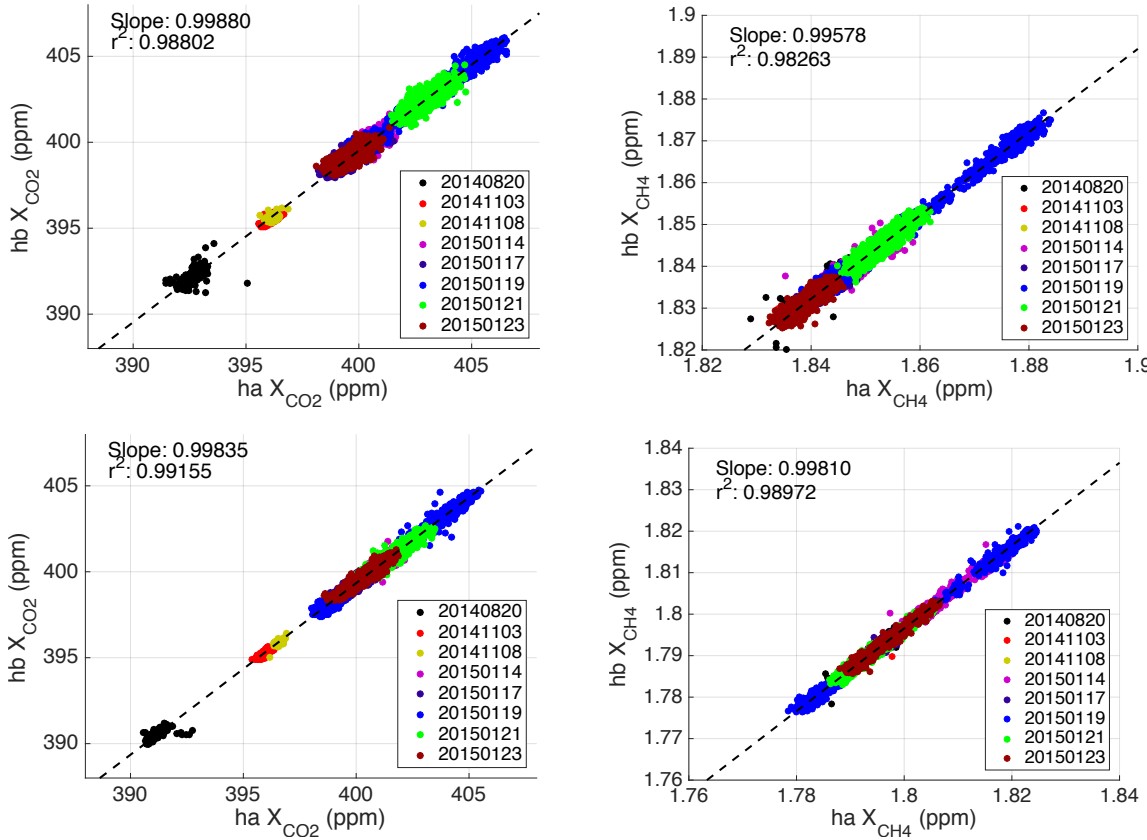

**Figure 2.** Scatter plots with the slopes representing $\overline{R_G}$ for different days using I2S/GFIT retrieval (top panels) and PROFFIT retrieval (bottom panels). January measurements are carried out in Pasadena, others in Boston. The first four days are before the field study, others are during the campaign.

## 4 Scientific Applications

### 4.1 Emission of an Area Source

We measured the column-averaged dry-air mole fractions $X_{CO_2}$ and $X_{CH_4}$ simultaneously at locations upwind and downwind of the dairy farms in Chino, California, for several days in January 2015. Field results for *ha, hb*, and *pl* are shown in Fig. 3.

Meteorological conditions were particularly favorable on 24 Jan. 2015, with consistent wind directions and wind speeds ($\sim$10 m s$^{-1}$) at both Chino airport (KCNO: 1 km northeast of the downwind station $ha$) and Ontario airport (KONT: 3 km north of the upwind station $hb$) (see Fig. 3 (c) panels 3 and 4). 5 minutes averaged wind information from Automated Surface Observing System (ASOS) is used.

The measured methane enhancement $\Delta X_{CH_4}$ was notably consistent at $\sim$2 ppb over 5 hours of measurement (Fig. 3 (c), Table 2), 10 times larger than our measurement precision using 5 min integration time (see Sec. 3.1). Times between 0.1 and 0.7 hours after solar noon were not taken into account, because transient peaks were measured at the upwind site. The transient peaks are also observable at the downwind site, but weaker due to dispersion. More discussions on the transient peaks can be found in Sec. 4.3.2.

Winds were more variable on 15 Jan. 2015 (Appendix F), with a consistent period of just $\sim$1 hour with relatively light wind ($\sim$2 m s$^{-1}$) at the two airports. The observed $\Delta X_{CH_4}$ was $\sim$10 ppb (Fig. 3 (d)), a factor of 5 larger than on the 24$^{th}$, showing inverse proportionality to the wind speed.

We use a simple column model (Jacob (1999)) and literature emission values to *estimate* $\Delta X_G$ for the dairy farms (area source) and to *verify* our measurements:

$$\Delta X_G = X_G^d - X_G^u = \frac{D}{\overline{U}} \times \frac{E_G}{\text{column}_{\text{dryair}}}, \tag{4}$$

where $E_G$ is the mean emission flux (unit: molec. m$^{-2}$ s$^{-1}$) along the line traversing the area source, and $D$ is the length of the transect. column$_{\text{dryair}}$ denotes the mean column number density of dry air. The frame of reference is the air column, which picks up the emissions of gas $G$ from the dairies as the air traverses the farms. The longer the air column travels in the emission field, the larger the difference between the column number densities of the downwind and upwind sites will become. $\Delta X_G$ is therefore proportional to the residence time $D/\overline{U}$ of the air column, and inversely proportional to the wind speed $\overline{U}$. This simple column model is applicable when the wind direction and speed are consistent across the area, and fluxes are uniform at plume scale.

Our model assumes that air parcels within the air column are transported with a mean velocity $\overline{U}$ in the horizontal direction, which can be estimated using real-time data for the wind speed at the surface. Using Reynolds' decomposition, the time series of horizontal ($u$) and vertical ($w$) wind speed are split into a mean part and a turbulent part, i.e.:

$$u(t) = \overline{u} + u_{\text{turb}}(t), \qquad\qquad w(t) = \overline{w} + w_{\text{turb}}(t), \tag{5}$$

$$\sigma_u = \sqrt{<u_{\text{turb}}^2>}, \qquad\qquad \sigma_w = \sqrt{<w_{\text{turb}}^2>}. \tag{6}$$

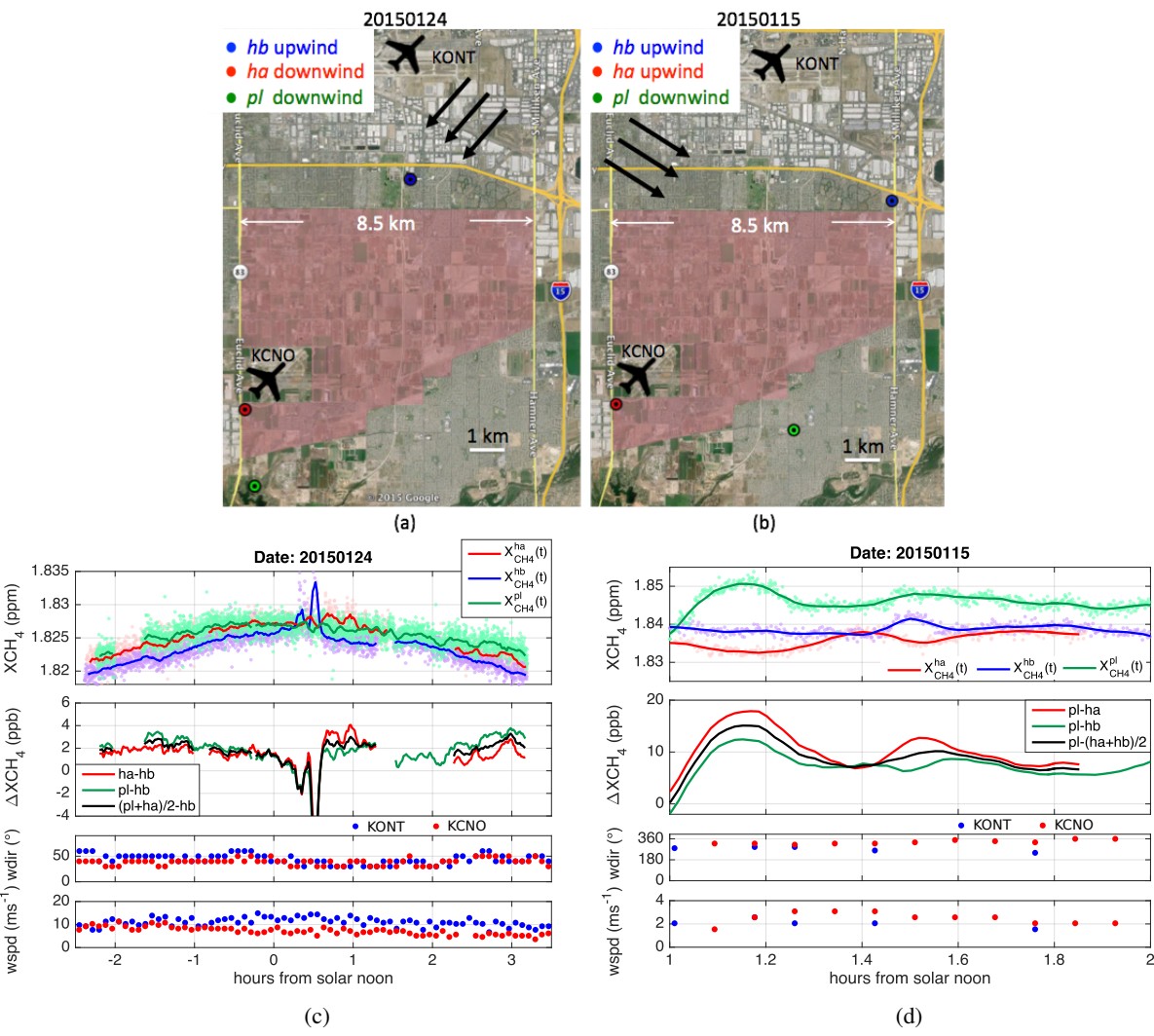

**Figure 3.** (a) and (b): locations of FTS stations and mean wind directions on 15 and 24 Jan., 2015. (c) and (d): column-averaged DMF measurements at three stations and downwind minus upwind differences (solar zenigh angle $\leq 70°$). On 24 Jan., $\Delta X_{CH_4}$ was steady at $\sim 2$ ppb most of the day, 10 times larger than our measurement precision; on 15 Jan., $\Delta X_{CH_4}(t)$ was $\sim 10$ ppb, about 5 times larger than on 24 Jan., showing inverse proportionality to the wind speed. Map provided by Google Earth, Image Landsat, Data SIO, NOAA, U.S. Navy, NGA, and GEBCO.

$\sigma_u$ and $\sigma_w$ are the standard deviations of the turbulent components. We assume the turbulence is horizontally homogeneous ($\sigma_u$ is independent of location) and isotropic ($\sigma_w = \sigma_u$), and that the mean vertical wind speed $\overline{w}$ is zero. Strictly speaking, $\overline{U}$ denotes the *mass-enhancement-weighted* wind velocity, i.e. $\overline{u}(z)$ weighted with the vertical distribution of the $CH_4$ molecules emitted from the dairies, denoted as $PDF_{\Delta CH_4}(z)$, i.e.

$$5 \quad \overline{U} = \int_0^\infty \overline{u}(z)\, PDF_{\Delta CH_4}(z)\, dz. \tag{7}$$

Note if $\overline{u}(z) = $ const., we have $\overline{U} = \overline{u}(z) = $ const., i.e. $\overline{U}$ is independent of the vertical distribution of the $CH_4$ molecules being added in the column. However, since the wind speed generally increases with altitude, $PDF_{\Delta CH_4}(z)$ needs to be considered for the estimate of $\overline{U}$.

We assume $\Delta CH_4$ is uniformly distributed up to a mixing height $z_{\mathrm{emiss}}$ and negligible above, then:

$$10 \quad PDF_{\Delta CH_4}(z) = \begin{cases} \dfrac{1}{z_{\mathrm{emiss}}}, & 0 \le z \le z_{\mathrm{emiss}}, \\ 0, & z > z_{\mathrm{emiss}}, \end{cases} \tag{8}$$

$$\overline{U} = \frac{1}{z_{\mathrm{emiss}}} \int_0^{z_{\mathrm{emiss}}} \overline{u}(z)\, dz. \tag{9}$$

We use a 2D random walk model (McCrea and Whipple (1940)) to estimate $z_{\mathrm{emiss}}$, the height to which $CH_4$ emissions are transported vertically by turbulent flow. The number of the random-walk steps $n$ is given by the ratio between the average transit time of the emission $\tau_{\mathrm{transit}}$ and the decorrelation time of the turbulent velocities $\tau_{\mathrm{eddy}}$, i.e.,

$$15 \quad n = \frac{\tau_{\mathrm{transit}}}{\tau_{\mathrm{eddy}}} = \frac{D \sigma_w}{2 \overline{u} \lambda}. \tag{10}$$

Assuming homogeneous emission, $\tau_{\mathrm{transit}}$ is approximately $D/2\overline{u}$, with $\overline{u}$ representing the mean speed at the surface. This also corresponds to the transit time of a particle emitted at the center of the field. $\tau_{\mathrm{eddy}}$ is given by $\lambda/\sigma_w$, where $\lambda$ denotes the average eddy scale.

On 24 Jan., the mean horizontal wind speed over the entire measurement time is 11.35 m s$^{-1}$ at KONT and 7.29 m s$^{-1}$
20 at KCNO with a standard deviation (1 $\sigma$) of 1.75 m s$^{-1}$ and 1.59 m s$^{-1}$, respectively. The wind directions are likewise very consistent over time, with a standard deviation of 8.9° (KONT) and 6.5° (KCNO). The wind speed at 10 m above ground level (agl) is assumed to be the average at the two airports over time, which gives $\overline{u}(10\,\mathrm{m}) = 9.3$ m s$^{-1}$ with fluctuations $\sigma_u(10\,\mathrm{m}) = \sigma_w(10\,\mathrm{m}) = 1.7$ m s$^{-1}$. Assuming an average eddy scale of 100 m, the expected value of the height to which $CH_4$ emissions rise is therefore:

$$25 \quad z_{\mathrm{emiss}} = \frac{\lambda \sqrt{n}}{\sqrt{2}} = \frac{1}{2} \sqrt{\frac{D \sigma_w(10\,\mathrm{m}) \lambda}{\overline{u}(10\,\mathrm{m})}} = \frac{1}{2} \sqrt{D\, I\, \lambda} \approx 200 \text{ m}. \tag{11}$$

According to Taylor's hypothesis (Taylor (1938)), the turbulence intensity $I = \sigma_u / \overline{u}$ should be constant, which indicates $z_{\mathrm{emiss}}$ does not depend on $\overline{u}$, but only on the eddy scale $\lambda$ and the turbulence intensity.

For determining $\overline{U}$ we need to consider the wind profile both in and above the surface layer. The wind follows a roughly logarithmic profile in the surface layer. At the middle portion of the PBL, the wind has typically constant direction and speed (Stull (1988)). Because the surface roughness information is not available, we use the power law to approximate the log wind profile in the surface layer and assume a constant horizontal wind speed above, i.e.,

$$\overline{u}(z) = \begin{cases} \overline{u}(10\text{ m})\left(\frac{z}{10\text{ m}}\right)^{\alpha}, & 0 \leq z \leq z_{\text{S}}, \\ \overline{u}(10\text{ m})\left(\frac{z_{\text{S}}}{10\text{ m}}\right)^{\alpha}, & z_{\text{S}} < z < z_{\text{PBL}}, \end{cases} \tag{12}$$

where $z_{\text{S}}$ and $z_{\text{PBL}}$ denote the depth of the surface layer and the PBL, respectively. The power law exponent $\alpha$ is approximately $1/7$ for neutral stability conditions (Hsu et al. (1994)).

Inserting Eq. 12 into Eq. 9, we obtain:

$$\overline{U} = \frac{\overline{u}(10\text{ m})}{z_{\text{emiss}}} \left( \int_{0}^{z_{\text{S}}} \left(\frac{z}{10\text{ m}}\right)^{\alpha} dz + \int_{z_{\text{S}}}^{z_{\text{emiss}}} \left(\frac{z_{\text{S}}}{10\text{ m}}\right)^{\alpha} dz \right)$$

$$= \overline{u}(10\text{ m})\left(\frac{z_{\text{S}}}{10\text{ m}}\right)^{\alpha} \left(1 - \frac{z_{\text{S}}}{z_{\text{emiss}}} \frac{\alpha}{\alpha+1}\right). \tag{13}$$

Varying $z_{\text{S}}$ in the range of 10 m to $z_{\text{emiss}}$, we obtain $\overline{U}$ in the range of 9.3 to 12.5 m s$^{-1}$, corresponding to an average of 10.9 m s$^{-1}$ $\pm15\%$. The lower bound is given by a constant wind speed starting from 10 m agl, and the upper bound assumes a wind profile power law up to the mixing height $z_{\text{emiss}}$.

Our wind model is consistent with the wind data profiles taken during aircraft taking off and landing at Ontario airport via Aircraft Communications Addressing and Reporting System (ACARS), and also agrees well with the Hybrid Single-Particle Lagrangian Integrated Trajectory model (HYSPLIT) simulations (Stein et al. (2015); Draxler and Hess (1998)), driven by the meteorological model "North American Mesoscale Forecast System (NAM)" with a horizontal resolution of 12 km. The comparisons can be found in Appendix E.

Oxygen column number density is determined as $4.493 \cdot 10^{28}$ molec. m$^{-2}$ $\pm 0.5\%$ (Appendix D), accounting for 20.95% of dry air. According to Eq. 4, the uncertainty in calculating $E_{\text{G}}$ from $\Delta X_{\text{G}}$ is the sum of the uncertainties in $\overline{U}$ (15%), $\Delta X_G$ (0.01% precision), and column$_{\text{dryair}}$ (0.5%), in total roughly 16%. Therefore, an emission estimate using differential column measurements is dominated by the uncertainty in the transport (i.e. $\overline{U}$), not the differential column measurements themselves.

Since the two spectrometers have identical optical setup, spectral resolution, and measuring geometry, their column averaging kernels are very similar, and happen to be close to one at all altitudes (Hedelius et al. (2016)). The uncertainty arising from the differences in averaging kernels is included in the uncertainty in $\Delta X_G$. The sensitivity of $X_{\text{CH}_4}$ on surface pressure inputs is discussed in Appendix C.

In Table 2, the time-averaged $\Delta X_{\text{CH}_4}$ and their corresponding emission numbers are listed. Measurements between 0.1 and 0.7 hours after solar noon are neglected due to a transient peak measured with $hb$ (Fig. 3).

We can compare our measurements to the value of $\Delta X_{\text{CH}_4}$ derived from literature annual mass emission rates $E_{\text{CH}_4,\text{annual}}$ for the dairy farms. Peischl et al. (2013) determined 28 Gg yr$^{-1}$ using bottom-up method accounting for enteric fermentation

| Configuration | $\Delta X_{CH_4}$ (ppb) | $E_{CH_4}$ [molec. m$^{-2}$ s$^{-1}$)] | $E_{CH_4,annual}$ [Gg yr$^{-1}$] |
|---|---|---|---|
| $ha$ - $hb$ | 1.8 | $5.38 \cdot 10^{17} (\pm 16\%)$ | $22.5\ (\pm 26\%)$ |
| $pl$ - $hb$ | 2.1 | $6.15 \cdot 10^{17} (\pm 16\%)$ | $25.8\ (\pm 26\%)$ |
| $(pl + ha)/2$ - $hb$ | 2.1 | $6.09 \cdot 10^{17} (\pm 16\%)$ | $25.5\ (\pm 26\%)$ |
| Peischl's bottom-up | $2.3\ (\pm 0.6)$ | | 28.0 |
| Peischl's top-down | $(4.0 \pm 1.0)\ (\pm 50\%)$ | | $49.0\ (\pm 50\%)$ |

**Table 2.** Time-averaged $\Delta X_{CH_4}$, using $ha$ or $pl$, or $ha$ and $pl$ as downwind stations on 24 Jan. 2015, and their corresponding emission numbers calculated using Eq. 4. $\Delta X_{CH_4}$ is rounded to one decimal place in the table, whereas for the calculation of $E_{CH_4}$ and $E_{CH_4,annual}$ all available digits are used. THe uncertainty of 16% is given by the uncertainties in $\overline{U}$, column$_{dryair}$, and $\Delta X_G$. Uncertainty in $E_{CH_4,annual}$ is 16% added with 10% uncertainty in the emission area. This table also displays the annual emission rates estimated by Peischl et al. (2013) using bottom-up and top-down methods. The corresponding column differences $\Delta X_{CH_4}$ and their uncertainties, as derived in Eq. 14, are summarized in the second column.

and dry manure management, and 49($\pm 50\%$) Gg yr$^{-1}$ using top-down method with aircraft-based mass balance approach during the CalNex field study. We assume a dairy area (area$_{emiss}$) of 50 km$^2 \pm 10\%$ and a constant emission rate across the farm throughout day and night, to convert $E_{CH_4,annual}$ (unit: Gg yr$^{-1}$) to $E_{CH_4}$ (unit: molec. m$^{-2}$ s$^{-1}$). The transect length $D$ is approximated with 8 km, which is the diameter of a circle with 50 km$^2$ area.

For 24 Jan. 2015,

$$\Delta X_{CH_4,Exp} = \frac{D}{\overline{U}} \times \frac{E_{CH_4,annual}(\text{Peischl's Number})}{m_{CH_4} \cdot \text{area}_{emiss} \cdot N_{s/year} \cdot \text{column}_{dryair}}$$
$$= \begin{cases} 2.3 \pm 0.6 \text{ ppb}, & \text{for 28 Gg yr}^{-1} \text{ (bottom-up estimate)}, \\ (4.0 \pm 1.0)(\pm 50\%) \text{ ppb}, & \text{for 49}(\pm 50\%) \text{ Gg yr}^{-1} \text{ (top-down estimate)}, \end{cases} \tag{14}$$

where $m_{CH_4}$ denotes the molecular mass of methane [g/molec.] and $N_{s/year}$ represents the number of seconds per year.

The observed $\Delta X_{CH_4}$, $\sim 2$ ppb (Fig. 3(c), Table 2), falls in the lower half of the range from Peischl. Our results, and
Peischl's top-down estimates, both represent just a few days of data. The difference with Peischl's results using the aircraft-based mass balance approach could be due to seasonal factors, activity levels at the farms, uncertainties in $\overline{U}$, as well as in background concentration and boundary layer height for the aircraft measurements (Cambaliza et al. (2014)), or model errors. Longer deployments with more ancillary data, such as wind profiles, would be needed to refine the result. Further studies using a WRF-LES model will be presented in Viatte et al. (2016). The differential column measurement using compact FTSs has
shown the capability to determine the emission flux when deployed across an area source such as Chino farms.

## 4.2 Source Characterization Using Ratios of Column Differences

Pasadena is a city within the South Coast air basin (SCB) with heterogeneous $CO_2$ and $CH_4$ emissions, from different source types such as transportation, electricity generation, industry, landfills, and gas leaks from natural gas delivery system. The ratio of column differences can be used to characterize regional emissions. For example, Wunch et al. (2009) measured diurnal

changes of $X_{CH_4}, X_{CO_2}$, and $X_{CO}$ (temporal difference), and used the $CO_2$ emission inventories from the California Air Resources Board (CARB) and EDGAR (Emission Database for Global Atmospheric Research) to estimate emissions of $CH_4$ and CO in the SCB.

We deployed two EM27/SUN spectrometers, $ha$ and $hb$, located north (34.2N, 118.13W, 557 m asl) and south (34.11N, 118.14W, 172 m asl) of Pasadena, on 27 Jan. 2015. We measured the column difference between $ha$ and $hb$ during the course

of day, i.e. $\Delta X_G(t) = X_G^{hb}(t) - X_G^{ha}(t)$, which is shown in Fig. 4 third panel.

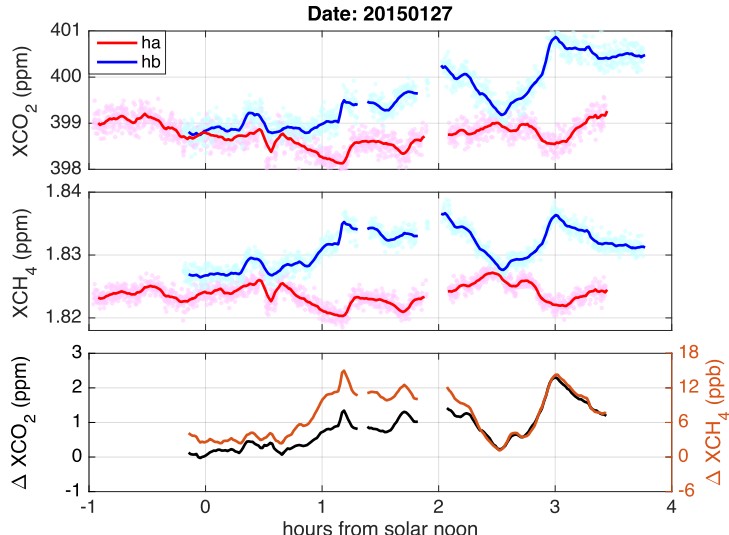

**Figure 4.** First and second panels: measured $X_{CO_2}$ and $X_{CH_4}$ north ($ha$) and south ($hb$) of Pasadena on 27 Jan. with 5 minutes averaging time. Third panel: $\Delta X_{CO_2}$ and $\Delta X_{CH_4}$ are temporally correlated and their ratio is determined as 7.8 ppb/ppm, shown in Fig. 5.

We determine the ratio of *spatial* column differences $\Delta X_{CH_4}/\Delta X_{CO_2}$ across Pasadena, by linear regression of $\Delta X_{CH_4}$ and $\Delta X_{CO_2}$ data using maximum likelihood estimation (York et al. (2004), see Fig. 5). The derived ratio over the course of the day ($7.8 \pm 0.1$ ppb/ppm) is consistent with the emission ratios determined by comparing the daily variations of $X_{CH_4}$ and $X_{CO_2}$ ($7.8 \pm 0.8$ ppb/ppm) reported at a TCCON station (Wunch et al. (2009)) located at JPL (34.2N, 118.2W, 390 m asl),

and likewise for the ratio of enhancements obtained by the CLARS-FTS ($7.28 \pm 0.09$ ppb/ppm, Wong et al. (2015)), which compared DMFs from diffuse solar reflectance off a spectralon plate at Mount Wilson (34.22N, 118.06W, 1670 m asl) with those from reflected sunlight from West Pasadena (34.17N, 118.17W).

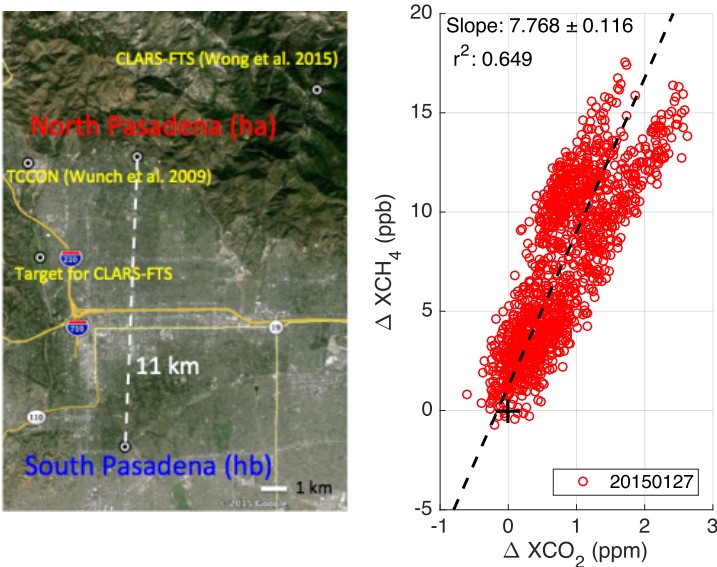

**Figure 5.** The derived ratio of column differences across Pasadena is consistent with Wunch et al. (2009) using TCCON DMF daily dynamics and very close to the excess ratio determined in Wong et al. (2015), which compared DMFs from diffuse solar reflectance off a spectralon plate at Mount Wilson with those from reflected sunlight from West Pasadena. Map provided by Google Earth, Image Landsat, Data SIO, NOAA, U.S. Navy, NGA, and GEBCO.

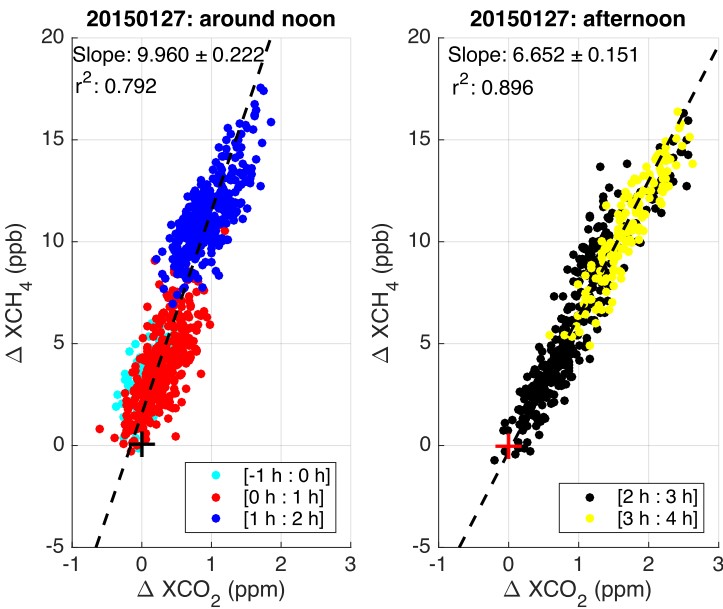

**Figure 6.** Column difference ratio measured across Pasadena colored by hours. Left figure shows the time period between 1 hour before solar noon and 2 hours after solar noon, and right figure shows the time period 2 to 4 hours after solar soon. The solar noon is about 12:05 local time. Both regression curves essentially pass through the origins that are shown as crosses.

By coloring the ratios of column differences per hour (Fig. 6), we observe a difference in the ratios between noon (9.96 $\pm$ 0.22 ppb/ppm) and afternoon time (6.65 $\pm$ 0.15 ppb/ppm), with both regression curves passing essentially through the origins. We determined a higher $\Delta X_{CO_2}/\Delta X_{CH_4}$ ratio in the afternoon than the noon time, which can be caused by more traffic emissions in the basin. The lagged cross covariance between $\Delta X_{CO_2}$ and $\Delta X_{CH_4}$ peaks at zero lag (Appendix G, Fig. 15, third panel), and the peak value is interestingly higher than the peak values of the cross covariance functions between $X_{CO_2}$ and $X_{CH_4}$ at individual sites (Fig. 15, first and second panels), which suggests the column difference is sensitive to the emissions between the two sites.

The Pasadena study confirms that sources of $CH_4$ are surprisingly large from SCB, as reported by previous papers using aircraft and TCCON data (Wunch et al. (2009); Wennberg et al. (2012); Peischl et al. (2013)). The capability of determining emission ratios using ratios of spatial column differences has been illustrated in this section.

## 4.3 Short-Term Variations

### 4.3.1 Side-by-side Measurements

We observed short-term variations in side by side measurements at Caltech and Harvard. These fluctuations are captured by both instruments simultaneously, representing geophysical phenomena, not noise as might be assumed. The high frequency temporal structure ($\sim$ 5-10 min Full Width at Half Maximum) can be caused by emissions not well mixed within the boundary layer ("plumes"), or by turbulence across the top of the ML, or by intrusions of a sea breeze front that introduces a different volume of air to the column, etc.

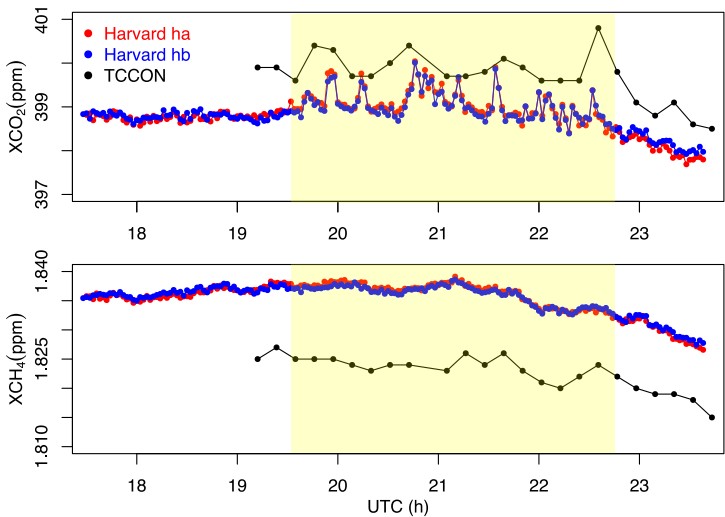

**Figure 7.** Side by side measurements on the roof of Caltech (17 Jan. 2015). Both EM27/SUN spectrometers (2 minutes block-average) captured short-term variations in the $X_{CO_2}$ signal ($\sim$ 1 ppm corresponds to 0.25% relative). The TCCON spectrometer does not resolve these short-term variation due to the low measurement rate.

In Fig. 7 we show, as an example, side by side measurements at Caltech. Short-term variations in $X_{CO_2}$ as large as 1 ppm are observed between 19:30 and 23:00 UTC. These features are only present in $X_{CO_2}$, not in $X_{CH_4}$. The wind directions during that time period were variable between south-southwest and northwest, indicating the short term variations are likely due to excess $CO_2$ emissions from a 12.5 MW combined heat and power (CHP) plant located ∼ 200 m to the south-southwest and/or

5   a solid oxide fuel cell ∼ 20 m to the northwest. Note that, because the co-located TCCON spectrometer samples at a lower rate, these variations are not well resolved in the TCCON data (Wennberg et al. (2014)).

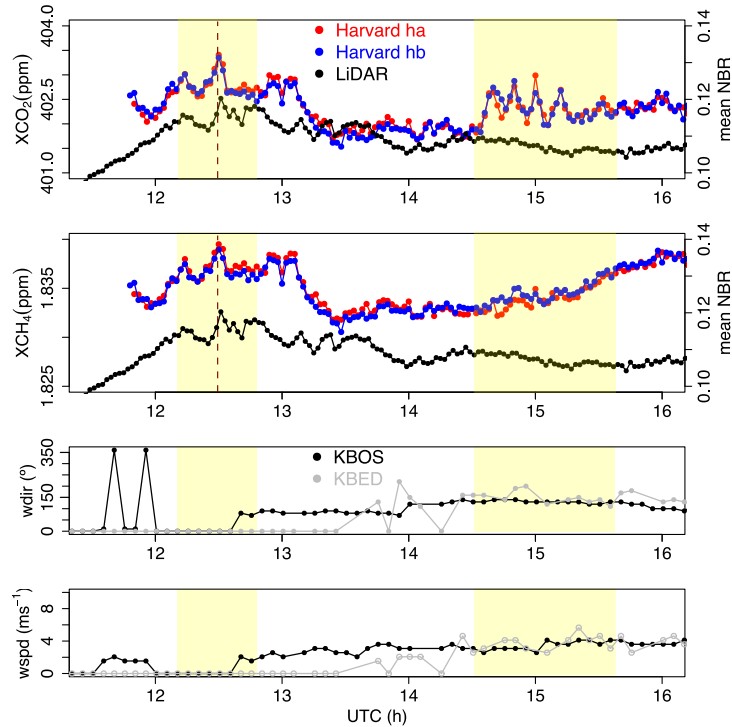

**Figure 8.** Short-term variations in $X_{CO_2}$ and $X_{CH_4}$ observed on the roof of Harvard Science Center by $ha$ and $hb$ (date: 16 April 2015; 2 min mean block average). A LiDAR metric of the thickness of the PBL (mean NRB 0-1.5 km agl) from a nearby site (Boston University, 3 km to the south-southeast of our site) is overlaid on top. The NRB signal is positively correlated with the short-term variations of the FTS measurements at 12-13 UTC.

Fig. 8 shows another example in Boston where $X_{CO_2}$ and $X_{CH_4}$ vary together by approximately the same relative amount (0.1-0.2%) at 12-13 UTC, also correlated with changes (∼ 10%) in the mean relative backscatter (NRB) measured from our LiDAR station 3 km away (Appendix H). The wind measurements at Boston Logan International Airport (KBOS), showing

10   easterly winds during that time period, differ from New Bedford Regional Airport (KBED), indicating a sea breeze event that likely generates wind shear and turbulence across the top of the ML. The depth of PBL undergoes short-term variations that are also visible in the LiDAR data at 12:30 UTC. In this case, the column-averaged DMFs vary because the proportion of PBL air in the whole column changes. Also the sea breeze circulation pushes a different volume of air through the column, which

could result in a sporadic jump of $X_{CO_2}$ and $X_{CH_4}$. The short-term variation of $X_{CO_2}$ at 14:30-15:30 UTC is not observed in $X_{CH_4}$ and the LiDAR data. It is probably caused by $CO_2$ plumes within the PBL, similar to what we observed at Caltech and shown in Fig. 7.

### 4.3.2 Transient Peak at Chino

Not only for side-by-side measurements, but also during the field measurement, short term peaks are observed, as mentioned in Sec. 4.1. Transient peaks are moving from the upwind to the downwind site: they are observable at upwind site $hb$ between 0.1 and 0.7 hours after solar noon, and at $ha$ between 0.5 and 1.1 hours after solar noon (Fig. 9). They are not observable at $pl$ site, probably because the plume is very narrow. Compared to the upwind peaks, the downwind peaks have a time shift, and are weaker and broader due to air dispersion. The peaks travelling from upwind to downwind site along the trajectory provide

a proof that the same air mass is sampled.

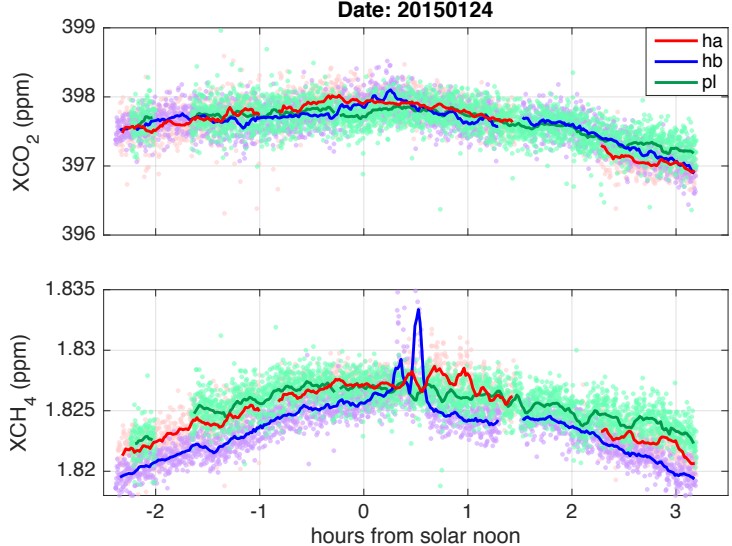

**Figure 9.** Observed column differences $\Delta X_{CO_2}$ (upper panel) and $\Delta X_{CH_4}$ (bottom panel) on 24 Jan. 2015, the transient peaks in $X_{CH_4}$ are not observed for $X_{CO_2}$.

    The transient peaks are not observed in $X_{CO_2}$, indicating they are not caused by passing clouds, or from a powerplant. They may come from natural gas leaks from the pipelines in the Chino area, with some evidents being reported by environmental defense fund (EDF (2016)). The transient peaks are removed from the column difference study (Sec. 4.1), because they are not associated with the local dairy farms.

The short-term variation helps us to understand the limitation of sampling using column measurements, which is relevant to the gradient determination and to satellite data. It is highly desirable to avoid aliasing these variations, and to characterize, model and/or measure the influences that cause these variations.

## 5    Conclusions

In this paper we demonstrated how to design observations of, and interpret, spatial gradients of column-averaged dry-air mole fractions for trace gases ($CO_2$, $CH_4$). We showed that the differential column methodology can be applied to the urban source problem and to other regional source-sink determinations.

We made extensive side-by-side measurements using two EM27/SUNs, in Boston and Pasadena, over many months. The differential system has a precision of $0.01\%$ for both $X_{CO_2}$ and $X_{CH_4}$ according to the Allan variance analysis when using an optimum integrating time of 10 min. The system is very stable in measuring column concentrations over time and after relocation across the continent.

We tested the gradient measurement and its sensitivity to emission sources, by measuring the downwind-minus-upwind column difference $\Delta X_{CH_4}$ across dairy farms in the Chino area. The ratio between the column difference and the measurement precision, i.e. Allan deviation was greater than 10, and the measured column gradient is inversely proportional to the wind speed. The derived emission numbers using a column model were consistent with the bottom-up source strength given by Peischl et al. (2013), and lie on the lower end of their top-down estimates using aircraft-based mass balance approach. Ratios of spatial column differences $\Delta X_{CH_4}/\Delta X_{CO_2}$ were measured across Pasadena within the South Coast air basin, with values consistent with emission ratios from the literature.

We observed significant short-term variations of $X_{CH_4}$ and $X_{CO_2}$, and showed that they are not noise or variation of optical path length, but represent atmospheric phenomena. These measurements provide useful information for measuring pollution plumes, turbulence across the top of the mixed layer, and transient peaks.

Overall, this paper helps establish a range of new applications for compact solar-tracking Fourier transform spectrometers, and shows the capability of differential column measurements for determining urban emissions. By accurately measuring the *differences* in the integrated column amounts across local and regional sources, we directly observe the mass loading of the atmosphere due to the influence of emissions in the intervening locale. The inference of the source strength is much more direct than inversion modeling using only surface concentrations, and less subject to errors associated with small-scale transport phenomena. The advent of compact, robust solar viewing spectrometers opens up myriad applications not hitherto pursued.

## 6    Data avalibility

The data for this study is available through the webpage:

https://dataverse.harvard.edu/dataset.xhtml?persistentId=doi:10.7910/DVN/J2YPX3

## Appendix A:  Instrument Line Function Parameters

The measured spectrum is a convolution between the atmospheric spectrum and instrument line shape in the frequency domain ILS($\nu$). In the ideal case, ILS($\nu$) is a delta function, which corresponds to a constant modulation efficiency for all optical path length differences. However, in practice, ILS($\nu$) is broader than a delta impulse, caused by the spectrometer's finite

optical pathlength, finite aperture size and also misalignment of the interferometer. The ILS in the interferogram domain can be approximated using a simple model that assumes a linear decay of the modulation efficiency with increasing optical path difference and a constant phase error (Hase et al. (1999)).

We estimated the ILS parameters of both spectrometers with an experimental setup, similar as described in Frey et al. (2015) and determined the modulation efficiency at maximum optical path difference ($OPD_{max}$) and phase error using the simple model implemented in the LINEFIT software (Hase et al. (1999)). Matlab scripts for automation purposes have been developed and can be obtained from the corresponding author.

Even though the measured ILS parameters are different for the two spectrometers due to the different internal alignment, the ILS of each single instrument is consistent over time and after relocation of the instrument across the contiguous US (see Table 3).

| Boston | | |
|---|---|---|
| Instrument | modulation efficiency at $OPD_{max}$ | phase error (rad) |
| $ha$ | 0.975 | $-3 \cdot 10^{-3}$ |
| $hb$ | 0.988 | $5 \cdot 10^{-3}$ |
| Pasadena | | |
| Instrument | modulation efficiency at $OPD_{max}$ | phase error (rad) |
| $ha$ | 0.974 | $-2 \cdot 10^{-3}$ |
| $hb$ | 0.990 | $4 \cdot 10^{-3}$ |

**Table 3.** Modulation efficiency and phase error determined for EM27/SUN $ha$ and $hb$ in Boston and Pasadena.

## Appendix B: Calibration factors for $ha$ and $pl$

Table 4 shows the calibration factors for $ha$ and $pl$ to match $hb$ measurements, determined by linear regressions of the side-by-side measurements on the roof.

| Instrument | $\overline{R_{CH_4}}$ | $\overline{R_{CO_2}}$ | $\overline{R_{O_2}}$ |
|---|---|---|---|
| $ha$ | 0.99578 | 0.99880 | 1.00846 |
| $pl$ | 1.00093 | 0.99930 | 0.99712 |

**Table 4.** Calibration factors for $ha$ and $pl$ to match $hb$, for $X_{CH_4}$, $X_{CO_2}$ and oxygen column number density measurements.

The linear models applied are:

$$X_{CH_4}^{hb} = X_{CH_4}^{ha} \cdot \overline{R_{CH_4}^{ha}},$$ (B1)

$$X_{CO_2}^{hb} = X_{CO_2}^{ha} \cdot \overline{R_{CO_2}^{ha}},$$ (B2)

$$\text{column}_{O_2}^{hb} = \text{column}_{O_2}^{ha} \cdot \overline{R_{O_2}^{ha}},$$ (B3)

$$X_{CH_4}^{hb} = X_{CH_4}^{pl} \cdot \overline{R_{CH_4}^{pl}},$$ (B1)

$$X_{CO_2}^{hb} = X_{CO_2}^{pl} \cdot \overline{R_{CO_2}^{pl}},$$ (B2)

$$\text{column}_{O_2}^{hb} = \text{column}_{O_2}^{pl} \cdot \overline{R_{O_2}^{pl}}.$$ (B3)

## 5 Appendix C: Retrieval Sensitivity to Pressure Inputs

Surface pressure $p_{\text{surf}}$ is a main input for the GFIT retrieval, to derive the site pressure altitude for each spectrum (Wunch et al. (2011)). Inaccurate pressure measurements will introduce errors in the computed widths of the gas absorption lines, i.e. pressure broadening, and therefore the fitted volume mixing ratio scale factors (Wunch et al. (2011)) will be inadequate, with a biased DMF as a result.

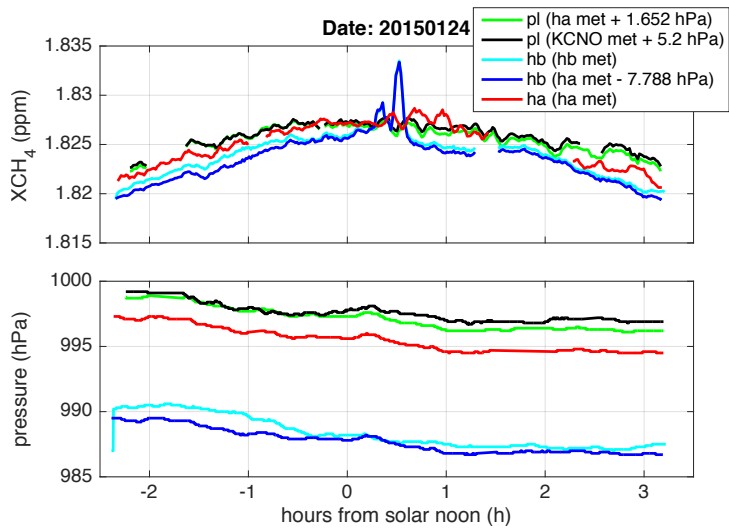

**Figure 10.** Surface pressure inputs (lower panel) and the corresponding $X_{CH_4}$ retrieval results averaged for 5 min (upper panel). For Fig. 3(c) and the simple column model calculations, $ha$, $hb$ and $pl$ retrievals with the surface pressure inputs based on $ha$ pressure measurements are used (red, blue and green curves).

On 24 Jan., the surface pressure measurement at $pl$ site failed, therefore we assess $p_{\text{surf}}^{pl}$ using pressure measurements at the closest FTS station $ha$ and the nearby airport KCNO. We assume hydrostatic equilibrium and a 1.18 hPa pressure difference per 10 m altitude difference. We derived $\Delta p_{\text{surf}}$ using the altitude difference between $pl$ and $ha$, as well as $pl$ and KCNO airport, and these two methods provide very similar results (Fig. 10). For the simple column model calculations, the retrieval with $p_{\text{surf}}^{pl}$ computed using $p_{\text{surf}}^{ha}$ and 1.652 hPa offset is used.

For consistency and a fair comparison with $ha$ and $pl$, $hb$ spectra are also retrieved with the surface pressure input calculated using $p_{\text{surf}}^{ha}$ and a negative 7.788 hPa offset, given by 66 m altitude difference. The strong wind could affect the pressure

measurement (Bernoulli's equation), which might be the reason for why $hb$ retrieval using its on-site pressure measurements slightly diverges from the result using the pressure data derived from $ha$ weather station data (blue curve in Fig. 10).

**Appendix D: Dry Air Column Measurements on 24 Jan. 2015**

Fig. 11 shows the $ha$, $hb$ and $pl$ measurements of oxygen column number densities. The deviations between three sites and the variations during the course of the day are given by the differences in atmospheric surface pressures and water column number densities. $ha$ and $pl$ measurements are scaled with the factors shown in Table 4 third column.

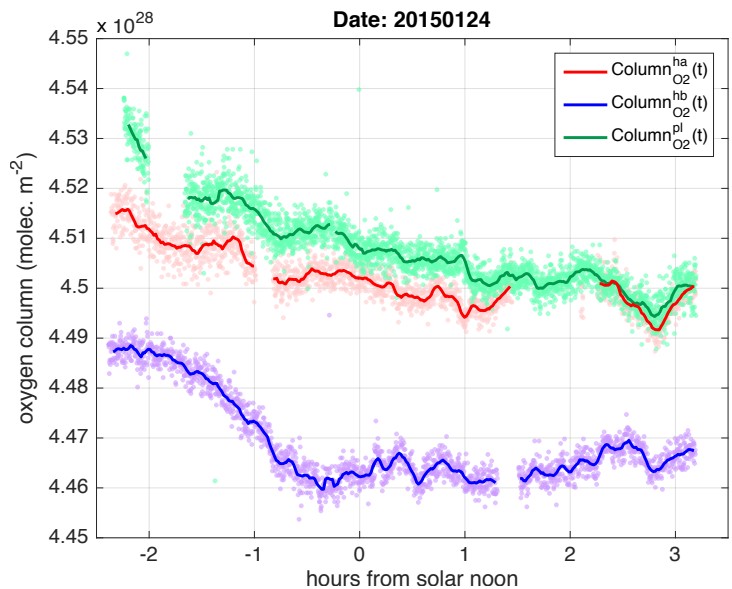

**Figure 11.** Oxygen column number densities measured by $ha$, $hb$ and $pl$ on 24 Jan. 2015.

According to Fig.11, the oxygen column number density over the dairy area is $4.493 \cdot 10^{28}$ molec. m$^{-2} \pm 0.5\%$, which is used to calculate column$_{\text{dryair}}$, the column number density of dry air. column$_{\text{dryair}}$ is needed in the column model (Eq. 4) for the emission estimates. The uncertainty is mainly associated with the differences in altitudes between sites.

**Appendix E: Validation of Wind Model**

As we have seen, the uncertainties in the wind speed estimates have a significant impact on the emission estimates. Hence, we check our wind model for plausibility in the following, by comparing it with ACARS profiles and HYSPLIT simulations.

## E1 ACARS Profile

In Fig. 12 we show the automated aircraft reports on profiles measured when taking off and landing at Ontario airport (MADIS ACARS profile data) and the automated surface observing system (ASOS) data, to examine the validity of the wind model described in Eq. 12. Two profiles were captured during the FTS measurement period, where only data above 2000 m agl are available, and two profiles after the measurement period. For plotting the model, we assume the surface layer height is $z_S = 100$ m and the power law exponent $\alpha = 1/7$. The potential temperature profiles during the FTS measurements and 2-3 hours after

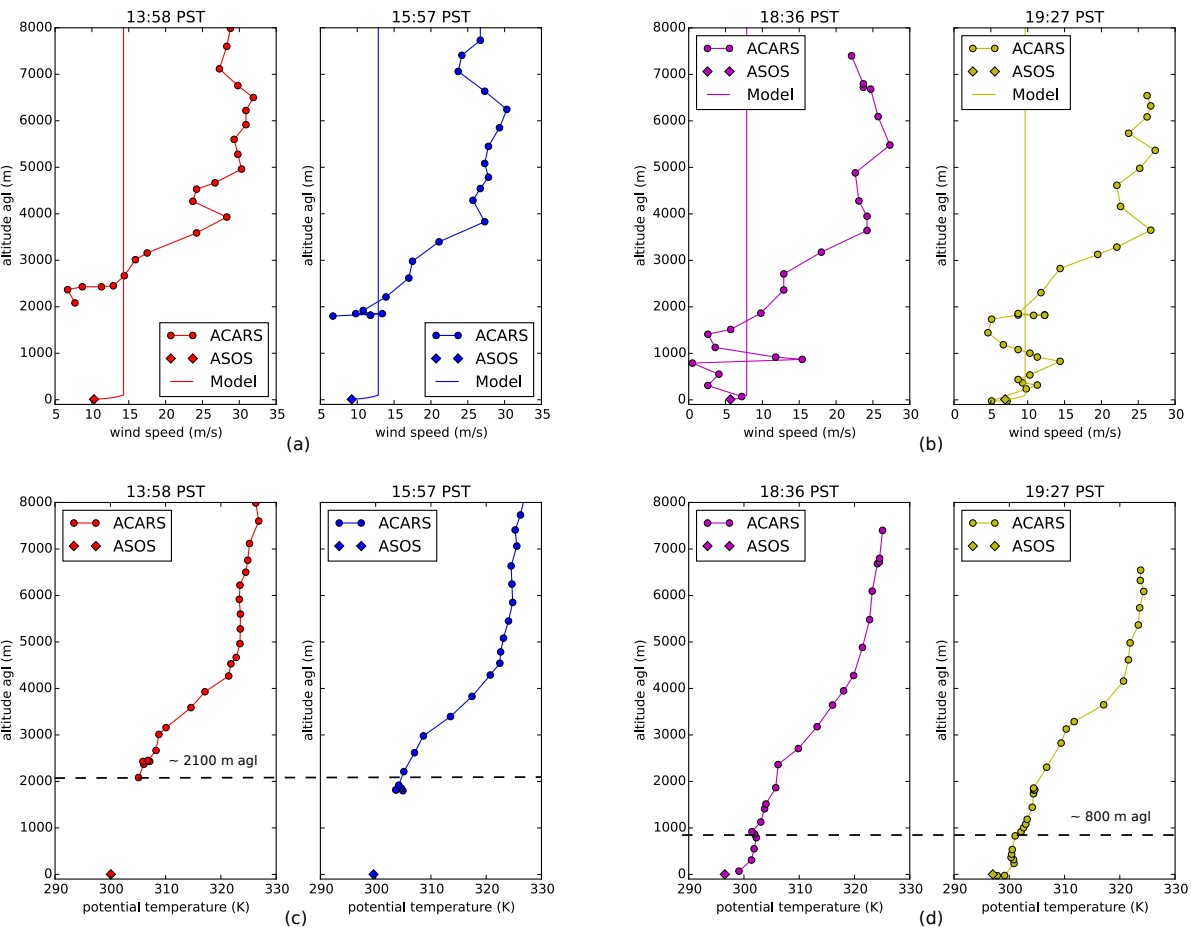

**Figure 12.** Vertical profile of the horizontal wind speed measurements and the calculated potential temperature profiles, at Ontario airport during (13:58 PST and 15:57 PST) and after the FTS measurements (18:36 PST and 19:27 PST).

are also shown in Fig. 12 (c) and (d). For the calculations we use the ACARS temperature profiles and the pressure profiles derived from the barometric formula, the ASOS sea level pressure data and a scale height of 7.4 km. Within the middle portion of the ML, the temperature profile follows adiabatic lapse rate, i.e. the potential temperature is nearly constant with height.

This behavior is observed between roughly 200 m and 800 m agl at 2-3 hours after the measurement period. During the FTS measurement period, the airplanes capture data above 2100 m agl where the adiabatic process is not observed. Therefore the PBL height is determined to be in the range of 800 to 2100 m agl. The surface layer is typically the bottom 10% of the PBL.

### E2    HYSPLIT Simulation

We use the Hybrid Single-Particle Lagrangian Integrated Trajectory (HYSPLIT) model to calculate the backward trajectories of tracers released from the FTS stations, starting at 20 UTC. We determine the wind speed from the travelled distance in one hour. To obtain a wind speed profile, multiple atitudes for the tracer release were chosen.

The simulated wind speeds are shown in Fig. 13. The blue curve represents the wind speed profile at the upwind site $hb$, and the red curve represents the wind speed profile at the downwind site $ha$. The black curve is the average of the two, which gives the mean wind profile. In addition, the wind profile used in our emission estimate (Eq. 12) is illustrated in Fig. 13 with the grey

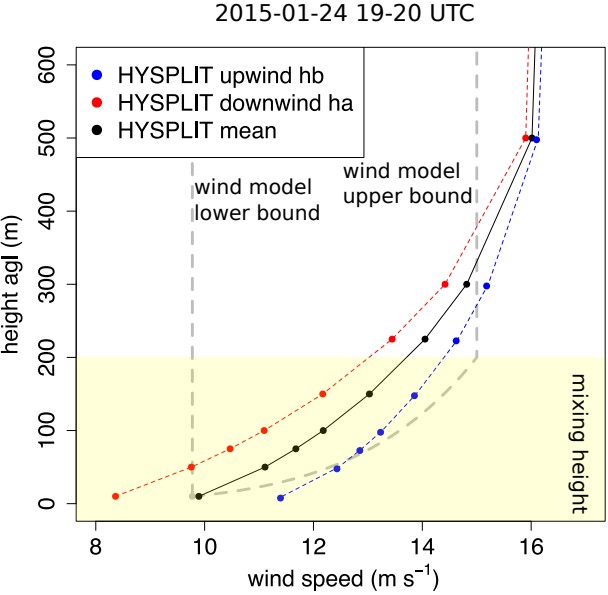

**Figure 13.** Comparison between our wind model and HYSPLIT simulations of wind speeds at different altitudes. The simulated mean wind profile (black curve) over Chino area is the averaged values of the profiles above the upwind site $hb$ (blue curve) and the downwind site $ha$ (red curve). The grey lines provide the range of our wind model.

lines. The lower bound is given by a constant wind speed starting from 10 m agl, and the upper bound assumes a wind profile power law up to the mixing height $z_{\mathrm{emiss}}$, which is determined as around 200 m (Eq. 11) using a random walk model. The wind speed at 10 m agl is assumed as the average of KCNO and KONT airports between 19 and 20 UTC (grey dot, 9.77 m s$^{-1}$), which is almost identical to the wind speed at 10 m agl determined using HYSPLIT mean (lowest black dot, 9.89 m s$^{-1}$).

In Fig. 13, the range given by our wind model covers the mean wind profile determined using the HYSPILT model (black curve). For the times between 19 and 20 UTC, $\overline{U}$ is determined as 11.6 m s$^{-1}$ ± 15% using our wind model (Eq. 13), which is consistent with the value of 12.1 m s$^{-1}$ obtained by averaging the HYSPILT wind profile vertically up to 200 m.

## Appendix F: Column Difference Measurement on 15 Jan. 2015

5    Fig. 14 shows the measurements of $X_{CH_4}$, $\Delta X_{CH_4}$, wind speeds, and wind directions for the entire day. For verifying the simple column model, we select the time period between 1 and 2 hours after solar noon with relatively consistent wind speeds and directions at KONT and KCNO. The data for the selected time window is shown in Fig. 3 (d).

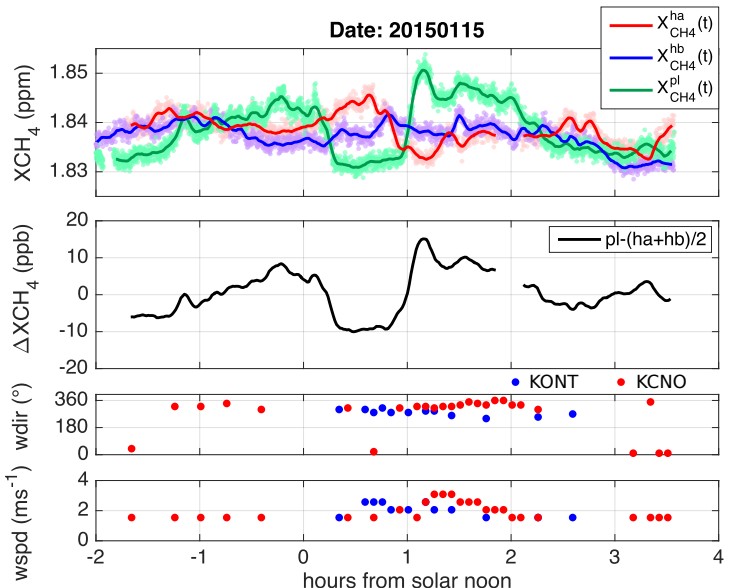

**Figure 14.** The whole day measurements on 15 Jan. 2015 for $ha$, $hb$ and $pl$. We neglect the wind measurements with zero wind speed. Gradient measurements between 1 and 2 hours after solar noon are selected (Fig. 3 (d)) because of the relatively consistent wind speeds and directions at KONT and KCNO.

## Appendix G: Cross covariance function between the $X_{CH_4}$ and $X_{CO_2}$ on 27 Jan. 2015

The lagged cross covariance between $X_{CH_4}$ and $X_{CO_2}$ for individual sites measured with $ha$ and $hb$, and between $\Delta X_{CH_4}$ and
10    $\Delta X_{CO_2}$ are shown in Fig. 15.

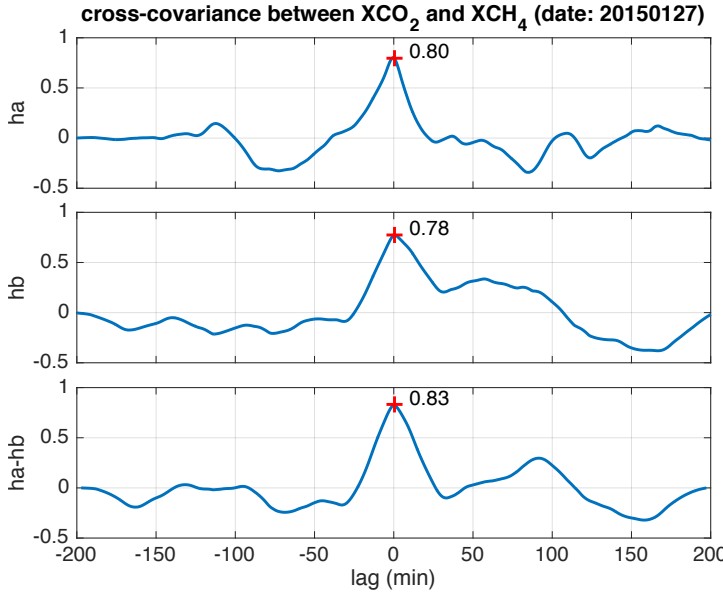

**Figure 15.** The cross covariance function (mean-removed cross correlation) between $X_{CH_4}$ and $X_{CO_2}$ for individual sites measured with *ha* and *hb* are shown in first and second panels. The lagged cross covariance between $\Delta X_{CH_4}$ and $\Delta X_{CO_2}$ (third panel) has a larger value at zero lag compared to the first and second panels.

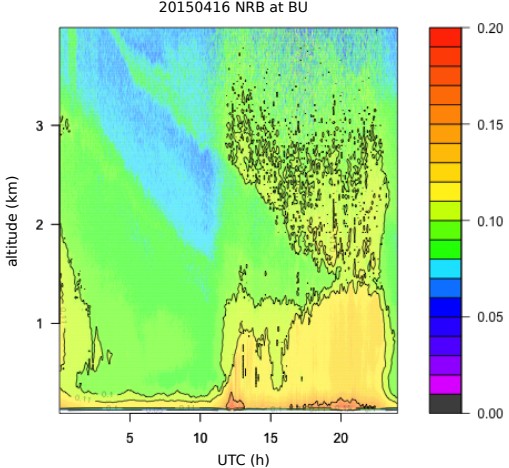

**Figure 16.** LiDAR measurement on the roof of Boston University on 20150416.

## Appendix H: LiDAR measurement in Boston

Fig. 16 shows the normalized relative backscatter (NRB) signal recorded using a Mini Micro Pulse LiDAR (MiniMPL from company Sigma Space) on the roof of the Boston University (BU) on 16 April 2015. We integrate the NRB signal vertically from 0 to 1.5 km to obtain the mean NRB. The time series of the mean NRB together with our FTS measurements are shown in Fig. 8.

*Acknowledgements.* We thank Bruce Daube, John Budney for the preparation of the measurement campaign in Chino and Pasadena, and for building the weather stations and the enclosures for the spectrometers. We thank Frank Hase for help with the PROFFIT retrieval software, Matthias Frey for instructions on the ILS measurements, and Matthäus Kiel for the Calpy software. We thank Yanina Barrera for the LiDAR data, and Frank Hase, Kelly Chance, Christoph Gerbig, Bruce Daube, John Budney, Bill Munger, Rachel Chang, and Kathryn McKain

5  for fruitful discussions. Funding for this study was provided by the National Science Foundation through Major Research Instrumentation Award 1337512 "Acquisition of Mesoscale Network of Surface Sensors and Solar-tracking Spectrometers". Jia Chen was partly supported by Technische Universität München - Institute for Advanced Study, funded by the German Excellence Initiative and the European Union Seventh Framework Programme under grant agreement n° 291763. Harrison Parker and Manvendra K. Dubey (Los Alamos National Laboratory) acknowledge NASA's Carbon Monitoring Program for funding the EM27/SUN application development. The authors gratefully acknowledge

10  the NOAA Air Resources Laboratory (ARL) for the provision of the HYSPLIT transport and dispersion model and/or READY website (http://www.ready.noaa.gov) used in this publication. The authors would also like to thank the anonymous reviewers for helpful comments.

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
