# Peer review of "Differential Column Measurements Using Compact Solar-Tracking Spectrometers"

_Atmospheric Chemistry and Physics, 2015_

## Referee Comment (RC1) · Anonymous Referee #1 · 23 Mar 2016

This paper describes some instrument characterisation and two applications of a recently commercially-available mobile Fourier transform spectrometer (Bruker EM27/SUN) coupled to a solar tracker for remote sensing of atmopsheric trace gases using the near infrared solar spectrum. The instrument is a low resolution mobile version of the FTS instruments used in the Total Carbon Column Observing Network (TCCON). The first part describes side-by-side measurements with three spectrometers and characterises the precision and stability amongst the three. The second part describes two applications of differential measurements when the spectrometers are located upwind and downwind of sources - a large dairy farm in California, and across an urban area in greater Los Angeles. The dairy farm is a nice example of exploiting the high repeatability and low bias of the instruments to detect small differences and spatial gradients.

Previous papers, eg by Gisi et al (2012), Frey et al., (2015) and Hedelius et al (2016, see references) have described similar characterisations and applications, but this does not take away from the usefulness of this paper, which extends rather than repeats those studies. The result is a useful addition to the characterisation of the precision, accuracy and stability of these spectrometers, which are finding many applications which complement TCCON. I recommend publication after addressing the comments listed below. The paper is rather technical in nature and does not contain a lot of new science, and therefore may be better suited to AMT than ACP.

General Comments:

In section 2 there is too little description of the basic solar FTS measurement - the reader is assumed to be familiar with solar FTS remote sesning and previous work. A summary of the measurement technique, referring back to TCCON and the previous papers on this instrument, would be useful for all readers, as it is it is directed only to those who are already involved in these measurements.

Section 3 is about precision and accuracy of the technique, but the authors use the term "precision" and other terms in incorrect ways. I recommend a reading of the IUPAC publication commonly known as GUM: "Evaluation of measurement data — Guide to the expression of uncertainty in measurement" by the Joint Committee for Guides in Metrology, JCGM 100-2008 (available from the BIPM website). "Precision" is a general term which is not defined for quantitative uncertainty assessment - quantities such as repeatability and reproducibility have specific, quantitative meanings and should be used in quantitative assessments of random uncertainty. Similarly there are more specific terms for "accuracy". In particular, in most cases in this paper, "precision" is used very loosly and mostly means "repeatability". Since the focus of section 2 is quantification of uncertainty, I recommend using correct terminology.

The authors use "gradient" throughout when they really mean "difference". A gradient is difference per unit length, eg 2 ppm km-1, not 2 ppm as used here. I found this

confusing when reading, and recommend that all instances of "gradient" be sought out and replaced with "difference" where appropriate.

Units: please quote units correctly, eg m s-1 not m/s, molec m-2 s-1, not molec/(m2s) as in Table 1. See the IUPAC "green book" (Cohen, et al. (2007). Quantities, Units and Symbols in Physical Chemistry. Cambridge, IUPAC, RSC Publishing) for authority on units.

Use of "%" and "‰": I found it very confusing to mix these two quantities, it is too easy not to notice ‰ and read it as %. I recommend using % throughout.

Specific and technical comments:

section 3.1. It is acceptable to use "precision" when speaking generally, but in cases when a value is assigned to an uncertainty, "precision" should be replaced by the appropriate specific quantity "repeatability" or "reproducibility". There are many instances, please search and replace.

P4 L10: "underestimate the true precision" really means repeatability and is ambiguous since high precision is a small number. Perhaps replace with "overestimate the true random uncertainty of the measurement."

P4 L21: Allan Deviation, not standard deviation.

P4 L28: "Allan standard deviation" is incorrect - replace with "Allan deviation", which is the square root of the Allan Variance,; they are not calculated in the same way as variance and standard deviation and should not be confused. There are many examples of "Allan standard deviation" throughout which should be replaced by "Allan deviation" = please search and replace (including figures, eg axis labels in Fig 1).

P5 L11 "System Robustness" "Robust" has a specific meaning in statistics, and since this is a statistical section, I would recommend System stability as a better title.

P11 L12: "Column gradient observations" - first of many examples where "difference"

or "differential" should replace "gradient"

P6 L5: Here "precision" is used when "accuracy" is meant - this sentence describes a systematic error.

P7 L23: The usual Reynolds notation is to use u' for the turbulent wind speed component, rather than uturb, so that u(t) = u(bar) + u'(t)

P10 L19: Please explain the transient peak around solar noon - what is it due to, and justification for its removal.

P11 L14: The meaning of this heading is quite unclear. I suggest replacing "gradient" with "difference" as commented earlier, and bring the second the paragraph beginning Pasadena ..." ahead of the first, so it comes first after the heading. Most of the confusion lies in the incorrect use of "gradient"

P12 Fig 4: The lowest panel would be very much improved if both the XCO2 and XCH4 axis had a common zero line.

P12 Fig 5: In the plot, cut off the negative axes at (-1,-5) to better utilise the space and avoid large empty area of the plot. Use the (0,0) axes rather than L and right axes, so the origin is clear.

P13 Fig 6: Same comment as Fig 5. The origin crosses can't be seen. The time periods are quite unclear - what are [-1 h : 0 h] etc, relative to what time? More detail in caption required.

P13 L1: please rephrase as " a lower DXCH4/DXCO2 ratio" since this is the way the plots are presented (with CH4 on the Y axis).

Appendix D: The plots of the O2 column are not particularly informative, since they reflect mostly pressure not spectroscopic retrieval. I suggest to replace these plots with the O2 column : pressure ratio (corrected to dry air) - this should be a constant with a known value related to 0.2095 mole fraction of O2 in air.

---

## Referee Comment (RC2) · Anonymous Referee #2 · 24 Mar 2016

General comments:

The authors present the use of compact Fourier spectrometers for differential column measurements to estimate source and sinks of CO2 and CH4. This paper characterizes the instrumentation and gives two examples of application: one example is the emission of dairy farms and the other one of an urban area. The paper demonstrates the stability of this kind of instrumentation and its usefulness for source and sinks strength estimates. Thereby it fully supports the findings of Gisi et al, 2012, Frey et al, 2015, and Hase et al., 2015, see reference list of the paper. Therefore it will stimulate further use of this promising multi purpose method.

The subject is appropriate for publication in ACP. It might also fit to AMT. The paper is well written and I recommend publication after minor revisions as listed below.

[Figure]

Specific comments:

- In Fig. 2 results of different days of observation are presented. To compare results obtained with different retrieval codes the same days should be presented for both cases. Side by side measurements were conducted over many months, but just a few days are presented in this paper.

- In the time series of the upwind site a transient peak is observed (Fig. 3). When the up- and downwind site are located along the trajectory a downwind peak should be present as well? Such peaks travelling from up- to downwind site may provide a proof of sampling the same air mass.

- Table 3 lists the calibration factors for the spectrometers. I recommend to include these factors for side by side measurements performed before and after the campaign. In order to show the stability of the instruments these factors obtained before and after the campaign should be discussed and compared with those presented by Frey et al., 2015.

---

## Author Comment (AC1) · 30 May 2016

**Response to Reviewer 2**

**Jia Chen**

Manuscript Title:

"Differential Column Measurements Using Compact Solar-Tracking Spectrometers"

We would like to thank reviewer 2 for carefully reading the paper and giving helpful comments.

Below, the reviewers' original text is included in typewriter font. The answers are highlighted in blue, sans-serif fonts.

In Fig. 2 results of different days of observation are presented. To compare results obtained with different retrieval codes the same days should be presented for both cases. Side by side measurements were conducted over many months, but just a few days are presented in this paper.

Thank you for this comment. Now the same days are presented for GFIT and PROFFIT to infer the mean calibration factors: three days in Boston and five days in Pasadena.

The side-by-side measurements are conducted since August 2014. For clarity reasons, we chose 8 days for deriving the calibration factors that are used for the field study.

The modified figures please find below:

Figure 1: Scatter plots with the slopes representing  $\overline{R_G}$  for different days using I2S/GFIT retrieval (top panels) and PROFFIT retrieval (bottom panels). January measurements are carried out in Pasadena, others in Boston. The first four days are before the field study, others are during the campaign.

In the time series of the upwind site a transient peak is observed (Fig. 3). When the up- and downwind sites are located along the trajectory a downwind peak should be present as well? Such peaks travelling from up- to downwind site may provide a proof of sampling the same air mass.

We thank the reviewer for the very useful hint. Yes, we think the transient peak is also observable at the downwind site, but much weaker probably due to dispersion. More discussions are added in Section 4.3.2:

**Transient Peak at Chino**

Not only for side-by-side measurements, but also in the field measurement, short term peaks are observed, as mentioned in Sec. 4.1. Transient peaks are moving from the upwind to the downwind site: they are observable at upwind site hb between 0.1 and 0.7 hours after solar noon, and at ha between 0.5 and 1.1 hours after solar noon (Fig. 2). They are not observable at pl site, probably because the plume is very narrow. Compared to the upwind peaks, the downwind peaks have a time shift, and are weaker and broader due to air dispersion. The peaks travelling from upwind to downwind site along the trajectory provide a proof that the same air mass is sampled.

---

## Author Comment (AC2) · 30 May 2016

The comment was uploaded in the form of a supplement: http://www.atmos-chem-phys-discuss.net/acp-2015-1058/acp-2015-1058-AC2-supplement.pdf

---

## Author Response (AR1)

**Response to Reviewers**

Manuscript Title:
"Differential Column Measurements Using Compact Solar-Tracking Spectrometers"

We would like to thank the reviewers for carefully reading the paper and giving helpful comments.

Below, the reviewers' original text is included in typewriter font. The answers are highlighted in blue, sans-serif fonts. After our responses to the reviewers, we have listed the addtional changes we have made to the manuscript.

**Reviewer 1**

**General Comments:**

In section 2 there is too little description of the basic solar FTS measurement - the reader is assumed to be familiar with solar FTS remote sensing and previous work. A summary of the measurement technique, referring back to TCCON and the previous papers on this instrument, would be useful for all readers, as it is it is directed only to those who are already involved in these measurements.

Thanks for this comment. We added one subsection (Sec. 2.1), providing background information about the solar-tracking FTS techniques and discussing existing networks. We added citations to NDACC (Hannigan (2011)), and to TCCON (Toon et al. (2009), Wunch et al. (2010), Wunch et al. (2011)). For EM27/SUN, Gisi et al. (2011), Gisi et al. (2012), Frey et al. (2015), Hase et al. (2015), Klappenbach et al. (2015), Hedelius et al. (2016) are cited. For the working principles of FTS, Davis et al. (2001) and Griffiths and De Haseth (2007) are referred. The added section please find on page 3 of the manuscript.

Section 3 is about precision and accuracy of the technique, but the authors use the term "precision" and other terms in incorrect ways. I recommend a reading of the IUPAC publication commonly known as GUM: "Evaluation of measurement data Guide to the expression of uncertainty in measurement" by the Joint Committee for Guides in Metrology, JCGM 100-2008 (available from the BIPM website). "Precision" is a general term, which is not defined for quantitative uncertainty assessment - quantities such as repeatability and reproducibility have specific, quantitative meanings and should be used in quantitative assessments of random uncertainty. Similarly there are more specific terms for "accuracy". In particular, in most cases in this paper, "precision" is used very loosely and mostly means "repeatability". Since the focus of section 2 is quantification of uncertainty, I recommend using correct terminology.

Thank you very much for this comment and providing the references. We looked at "JCGM 200:2008 International vocabulary of metrology  Basic and general concepts and associated terms (VIM)", which defines "measurement precision" as the closeness of agreement between indications or measured quantity values obtained by replicate measurements on the same or similar objects under specified conditions, and explicitly says that "Measurement precision is used to define measurement repeatability, intermediate measurement precision, and measurement reproducibility."

In the Wikipedia article Accuracy and precision, "precision" is described as: "The precision of a measurement system, related to reproducibility and repeatability, is the degree to which repeated measurements under unchanged conditions show the same results". According to ISO 5725-1, "precision" is the closeness of agreement among a set of results.

Therefore, we would like to keep the terminology "precision", which is also used in the field of laser spectroscopy for quantifying the measurement repeatability.

The authors use "gradient" throughout when they really mean "difference". A gradient is difference per unit length, eg 2 ppm km-1, not 2 ppm as used here. I found this confusing when reading, and recommend that all instances of "gradient" be sought out and replaced with "difference" where appropriate.

Thank you for pointing out this problem. We fully agree and changed the document accordingly. Note, that there are some occurrences of "gradient" in the document, where we actually mean difference per unit length, and thus kept the expression.

Units: please quote units correctly, eg m s-1 not m/s, molec m-2 s-1, not molec/(m2s) as in Table 1. See the IUPAC "green book" (Cohen, et al. (2007)). Quantities, Units and Symbols in Physical Chemistry. Cambridge, IUPAC, RSC Publishing) for authority on units.

We changed the units written as the product of units, without any multiplication sign and left one space between the unit symbols.

Use of "%" and "‰"- I found it very confusing to mix these two quantities, it is too easy not to notice "‰" and read it as %. I recommend using % throughout.

Thanks for this note, we changed all the "‰" to "%".

**Specific and technical comments:**

Section 3.1. It is acceptable to use "precision" when speaking generally, but in cases when a value is assigned to an uncertainty, "precision" should be replaced by the appropriate specific quantity "repeatability" or "reproducibility". There are many instances, please search and replace P4 L10: "underestimate the true precision" really means repeatability and is ambiguous since high precision is a small number. Perhaps replace with "overestimate the true random uncertainty of the measurement."

Thank you very much for this comment. As for our answer to the use of "precision", please see our reply to your general comment.
We changed "underestimate the true precision" to "overestimate the true random uncertainty of the measurement."

P4 L21: Allan Deviation, not standard deviation.
P4 L28: "Allan standard deviation" is incorrect - replace with "Allan deviation", which is the square root of the Allan Variance; they are not calculated in the same way as variance and standard deviation and should not be confused. There are many examples of "Allan standard deviation" throughout which should be replaced by "Allan deviation" = please search and replace (including figures, eg axis labels in Fig 1).

Thank you for pointing this out. We replaced "Allan standard deviation" with "Allan deviation" throughout the document.

P5 L11 "System Robustness" "Robust" has a specific meaning in statistics, and since this is a statistical section, I would recommend System stability as a better title.

We changed "robustness" to "stability", thanks!

P11 L12: "Column gradient observations" - first of many examples where "difference" or "differential" should replace "gradient"

We changed "column gradient observations" to "differential column observations".

P6 L5: Here "precision" is used when "accuracy" is meant - this sentence describes a systematic error.

Thanks. The previous sentence was misleading, we changed the sentence to "Retrievals for ha have been scaled with RG for the Allan analysis (Sec. 3.1) "

P7 L23: The usual Reynolds notation is to use u' for the turbulent wind speed component, rather than uturb, so that u(t) = u(bar) + u'(t)

Thanks for pointing this out. We would prefer to use $u_\mathrm{turb}$, because u'(t) could be mis-understood as the first derivative of the signal u(t). Hence, $u_\mathrm{turb}$ might be less prone to misunderstanding, especially, for readers from other disciplines.

P10 L19: Please explain the transient peak around solar noon – what is it due to, and justification for its removal.

We added section 4.3.2 "Transient Peak at Chino" to discuss the possible cause of the transient peak. The transient peaks are removed for calculating the column difference, since they are not associated with the local dairy farms.

P11 L14: The meaning of this heading is quite unclear. I suggest replacing "gradient" with "difference" as commented earlier, and bring the second paragraph beginning Pasadena ..." ahead of the first, so it comes first after the heading. Most of the confusion lies in the incorrect use of "gradient".

Thanks! We changed the title, and changed "gradient" to "difference". Also we changed the order of the first two paragraphs and hopefully the content is more clear now.

P12 Fig 4: The lowest panel would be very much improved if both the XCO2 and XCH4 axis had a common zero line.

Thanks much for this suggestion! Now the two y-axis share a common zero line in Fig. 4.

P12 Fig 5: In the plot, cut off the negative axes at (-1,-5) to better utilize the space and avoid large empty area of the plot. Use the (0,0) axes rather than L and right axes, so the origin is clear.

Thanks. We cut off the negative axes, and we use "+" to illustrate (0,0).

P13 Fig 6: Same comment as Fig 5. The origin crosses cant be seen. The time periods are quite unclear – what are [-1 h : 0 h] etc, relative to what time? More detail in caption required.

Thank you for the question and the suggestion. We changed the caption as "Left figure shows the time period between 1 hour before solar noon and 2 hours after solar noon, and right plot shows the time period 2 to 4 hours after solar soon."

P13 L1: please rephrase as " a lower DXCH4/DXCO2 ratio" since this is the way the plots are presented (with CH4 on the Y axis).

Done.

Appendix D: The plots of the O2 column are not particularly informative, since they reflect mostly pressure not spectroscopic retrieval. I suggest to replace these plots with the O2 column : pressure ratio (corrected to dry air) – this should be a constant with a known value related to 0.2095 mole fraction of O2 in air.

Thanks for this suggestion. Here the $O_2$ column are shown for determining column dry air and its uncertainty. We added one paragraph in Appendix D to clarify the purpose:
"According to Fig. 11, the oxygen column number density over the dairy area is $4.493 \cdot 10^{28}$ molec. m$^{-2} \pm 0.5\%$, which is used to calculate column$_\mathrm{dryair}$, the column number density of dry air. column$_\mathrm{dryair}$ is needed in the column model (Eq. 2) for the emission estimates. The uncertainty is mainly associated with the differences in altitudes between sites."

**Reviewer 2**

In Fig. 2 results of different days of observation are presented. To compare results obtained
with different retrieval codes the same days should be presented for both cases. Side by side
measurements were conducted over many months, but just a few days are presented in this paper.

Thank you for this comment. Now the same days are presented for GFIT and PROFFIT to infer the mean calibration factors: three days in Boston and five days in Pasadena.

The side-by-side measurements are conducted since August 2014. For clarity reasons, we chose 8 days for deriving the calibration factors that are used for the field study.

The modified figures please find on page 7 of the manuscript.

In the time series of the upwind site a transient peak is observed (Fig. 3). When the up- and
downwind sites are located along the trajectory a downwind peak should be present as well?
Such peaks travelling from up- to downwind site may provide a proof of sampling the same air
mass.

We thank the reviewer for the very useful hint. Yes, we think the transient peaks are also observable at the downwind site, but much weaker probably due to dispersion. More discussions are added in section 4.3.2.

Table 3 lists the calibration factors for the spectrometers. I recommend to include these
factors for side by side measurements performed before and after the campaign. In order to
show the stability of the instruments these factors obtained before and after the campaign
should be discussed and compared with those presented by Frey et al., 2015.

Thanks for this comment. We add Table 1 to the paper and list the calibration factors before and during the campaign. After the transport after the campaign, one screw for the flipping mirror was loose for ha and we opened the instrument to fix it, which might have changed the instrument behavior and resulted in a slightly different calibration factor. Since it is a technical problem, not representative for the overall behavior of EM27/SUN and differential column measurements, we decided to not report the calibration factors after the campaign in the paper.

The added table please find below:

|        | $\overline{R_{CH_4}}$ | | $\overline{R_{CO_2}}$ | |
|--------|---------|---------|---------|---------|
|        | GFIT    | PROFFIT | GFIT    | PROFFIT |
| Before | 0.99574 | 0.99813 | 0.99877 | 0.99838 |
| During | 0.99580 | 0.99809 | 0.99881 | 0.99834 |
| Both   | 0.99578 | 0.99810 | 0.99880 | 0.99835 |

Table 1: Calibration factors $\overline{R_G}$ for $X_{CH_4}$ and $X_{CO_2}$ before and during the field campaign, determined by forcing linear regression line go through zero. $\overline{R_G}$, determined using all data, are provided in the last row and used for the field study.

**Additional Changes**

1. We rephrased the first sentence in the introduction.

2. We changed the term "scaling factor" to "calibration factor".

3. We added a sentence "5 minutes averaged wind information from Automated Surface Observing System (ASOS) is used." in Sec. 4.1 to indicate where the wind data are coming from.

4. We expand Table 2 with Peischl's number for an easy reading and comparing.

5. We modified the title of Appendix C as "Retrieval Sensitivity to Pressure Inputs"

6. We changed the title of Appendix E to "Validation of Wind Model", and added Appendix E2 for validation with HSPLIT simulations, and added acknowlegment for HYSPILT,

7. We changed the order of Appendix E and Appendix F.

8. We thank the reviewers for helpful comments.

9. In some sentences the grammar was off, we corrected them.

**References**

[revised manuscript text omitted]